# Critical Southern Ocean climate model biases traced to atmospheric model cloud errors

Patrick Hyder[1], John M. Edwards[1], Richard P. Allan [2,3], Helene T. Hewitt[1], Thomas J. Bracegirdle[4], Jonathan M. Gregory[1,5], Richard A. Wood[1], Andrew J. S. Meijers[4], Jane Mulcahy[1], Paul Field[1,6], Kalli Furtado[1], Alejandro Bodas-Salcedo[1], Keith D. Williams[1], Dan Copsey[1], Simon A. Josey[7], Chunlei Liu [2,3], Chris D. Roberts[1], Claudio Sanchez[1], Jeff Ridley[1], Livia Thorpe[1], Steven C. Hardiman[1], Michael Mayer[8], David I. Berry[7] & Stephen E. Belcher[1]

The Southern Ocean is a pivotal component of the global climate system yet it is poorly represented in climate models, with significant biases in upper-ocean temperatures, clouds and winds. Combining Atmospheric and Coupled Model Inter-comparison Project (AMIP5/CMIP5) simulations, with observations and equilibrium heat budget theory, we show that across the CMIP5 ensemble variations in sea surface temperature biases in the 40–60°S Southern Ocean are primarily caused by AMIP5 atmospheric model net surface flux bias variations, linked to cloud-related short-wave errors. Equilibration of the biases involves local coupled sea surface temperature bias feedbacks onto the surface heat flux components. In combination with wind feedbacks, these biases adversely modify upper-ocean thermal structure. Most AMIP5 atmospheric models that exhibit small net heat flux biases appear to achieve this through compensating errors. We demonstrate that targeted developments to cloud-related parameterisations provide a route to better represent the Southern Ocean in climate models and projections.

[1] Met Office Hadley Centre, FitzRoy Road, Exeter, UK. [2] Department of Meteorology, University of Reading, PO Box 243, Whiteknights Campus Earley Gate, Reading RG6 6BB, UK. [3] National Centre for Earth Observations (NCEO), University of Reading, PO Box 243, Whiteknights Campus Earley Gate, Reading RG6 6BB, UK. [4] British Antarctic Survey, High Cross, Madingley Road, Cambridge CB3 0ET, UK. [5] National Centre for Atmospheric Sciences (NCAS), University of Reading, PO Box 243, Whiteknights Campus Earley Gate, Reading RG6 6BB, UK. [6] School of Earth and Environment, University of Leeds, Leeds LS2 9JT, UK. [7] National Oceanography Centre, European Way, Southampton SO14 3ZH, UK. [8] Department of Meteorology and Geophysics, University of Vienna, Althanstraße 14, Vienna 1090, Austria. Correspondence and requests for materials should be addressed to P.H. (email: patrick.hyder@metoffice.gov.uk)

The Southern Ocean plays an important role in global ocean heat and anthropogenic carbon uptake[1–4]. For example, recent climate model experiments suggest that 75 ± 22% of ocean heat uptake and 43 ± 3% of ocean anthropogenic carbon uptake over the historical period occurs south of 30°S[5]. The Southern Ocean also influences climate sensitivity[6], the Meridional Overturning Circulation (MOC)[7], water mass formation[7], sea level through basal melt of ice shelves[8], nutrient cycling[9] and the Inter-tropical Convergence Zone (ITCZ) position[10]. It has a persistent dynamical connection from the stratosphere to the deep ocean[11], involving clouds, air-temperatures, winds, surface heat fluxes, sea surface temperatures (SST), and ocean thermal structure leading to interrelated coupled model biases in all of these quantities[12,13].

Most coupled climate models have substantial warm biases in Southern Ocean SST[13,14] which have been linked to deficiencies in cloud processes, including a lack of reflective super-cooled liquid water and excessive downward surface short-wave radiation[15–19]. Errors in CMIP5 simulations of historical high-latitude southern hemisphere surface climate[20] and sea-ice[21] variability and trends could also be related to these deficiencies. In combination, these Southern Ocean errors are expected to contribute to uncertainties in regional climate projections that hinder our ability to cost-effectively manage climate change impacts[22,23]. Improved understanding of the causes and consequences of these biases is therefore urgently needed.

To address this requirement, a novel interpretive framework is presented to investigate the hypothesis that stand-alone atmospheric model surface flux errors cause coupled model SST biases. Analyses of CMIP5[24] and AMIP5[24] model biases are combined within a theoretical framework. The AMIP5 experiment uses the same atmospheric models employed in coupled CMIP5 simulations, but with a prescribed lower boundary condition of observed SST[24]. In AMIP5, the atmospheric state and surface heat fluxes are not influenced by coupled feedbacks caused by SST biases. Analysing CMIP5 and AMIP5 in combination therefore allows us to separate the influences of atmospheric model errors and coupled feedbacks, which were not separable in previous analyses that examined only CMIP5 simulations[17]. Equilibrium mixed layer heat budget theory is used to predict relationships between coupled model SST biases and atmospheric/coupled model net surface flux biases. The same relationships are then evaluated using linear regression analyses across the AMIP5/CMIP5 ensemble for the Southern Ocean 40–60°S region, followed by similar analyses for individual surface heat flux component biases. The impacts of AMIP5 net flux and CMIP5 SST biases on CMIP5 coupled model wind and ocean thermal structure biases are also investigated. Finally, new observational estimates are employed to investigate estimated simulated heat flux component biases for AMIP5 and a hierarchy of Hadley Centre atmospheric models, spanning over 20 years of model development.

The AMIP5/CMIP5 ensemble results, in combination with theoretical expectations, suggest that AMIP5 surface net flux bias variations are the dominant cause of variations in CMIP5 SST biases. The main cause of AMIP5 net flux bias variations is found to be short-wave radiation bias variations, which are known to be related to cloud deficiencies[15]. AMIP5 net flux and associated CMIP5 SST bias variations are also shown to impact on wind feedbacks and ocean thermal structure. Most AMIP5 models with small net flux biases appear to achieve this by error cancellation between flux components. Considerable improvements in atmospheric model surface flux component fidelity and coupled model SST biases in more recent Hadley Centre models demonstrate a route to improve coupled climate models.

## Results

**Observational products and their uncertainties**. Direct observations of ocean surface air-sea fluxes are extremely sparse, particularly for the Southern Ocean[25]. In consequence, large uncertainties in conventional observational estimates of surface heat fluxes hinder evaluation of the simulated surface energy budgets[18,25,26]. We therefore estimate net downward surface fluxes[27] from Top-Of-Atmosphere (TOA) satellite observations[28] and ERA-Interim[29] re-analysis energy divergences, assuming atmospheric column energy conservation[30]. This approach considerably constrains estimates of net surface flux derived from re-analyses[27] (see Methods and Supplementary Figure 1). We use CERES EBAF (Clouds and the Earth's Radiant Energy System Energy Balanced and Filled)[31] observational estimates of surface net downward short-wave radiation, long-wave radiation and total radiative (short-wave plus long-wave) fluxes. Net downward surface total turbulent fluxes, the sum of latent and sensible heat fluxes, are estimated as residuals, by subtracting CERES estimated surface total radiative fluxes from our observations-based estimated surface net fluxes. Adopting this approach avoids the use of turbulent flux products, which have particularly large errors[26]. For ocean temperature, including SST, and 10 m winds we employ observational estimates from EN4[32] and the ERA-Interim re-analysis[29], respectively.

We estimate multi-annual mean AMIP5/CMIP5 biases by subtracting observational estimates from the corresponding simulated quantities. Estimates of individual AMIP5/CMIP5 model biases include a contribution from observational errors. Since accurate quantitative observational error estimates are not currently available for our regional and temporal averages we derive basic indicative error estimates from differences between products (see Methods). Observational uncertainty estimates are quoted whenever observational errors could influence results. Overall, however, estimated observational errors are generally smaller than the substantial model biases, providing confidence in our inferences.

For linear regression analyses, unless otherwise stated, we consider area-weighted biases for the Southern Ocean 40–60°S region for 18 consistent AMIP5/CMIP5 models (see Methods and Supplementary Table 1 for model details). Most correlation values are presented together with regression analysis slopes in brackets or tables. We therefore generally adopt the terminology 'a correlation of Y on X' simply to indicate the direction of the associated regression analysis (although the correlation value is between X and Y).

It is important to emphasize that our regression correlations and slope results are not affected by any observational errors (since for each parameter the observational error will be identical across all the models). Correlations ($r$), regression slopes ($S$) and $p$ values ($p$) relate the variations in one parameter to those of another and are therefore independent of any reference employed. Values for our AMIP5/CMIP5 bias analyses would therefore be identical if we were to use different observational products to estimate biases; any individual model as a reference; or the actual parameters rather than estimated parameter biases. We do not interpret regression intercepts (which do depend on the observational errors). To indicate structural errors common to all models in regression figures we plot the estimated multi-model mean parameter biases together with their observational uncertainty.

**An interpretive framework for causes of SST biases**. A very strong correlation of 0.84 is evident between CMIP5 historical experiment[24] SST biases and AMIP5 net downwards surface flux biases (Table 1; Fig. 1). It is surprising that a characteristic of stand-alone atmospheric models with prescribed SST should be so closely associated with SST biases in coupled models spun up for many hundreds of years. To interpret this interesting result we

**Table 1 Regression and correlation analyses results across the AMIP5/CMIP5 ensemble**

| Regression relationship (40–60°S area mean biases unless stated) | Correlation (r) | Regression slope (S) (included if relevant) | p Value | No. of models |
|---|---|---|---|---|
| **Theory and AMIP5/CMIP5 ensemble results** | | | | |
| CMIP5-AMIP5 net flux on SST (R1) | −0.66 | −5.5 ± 1.6 $Wm^{-2}K^{-1}$ | 2.8E-3 | 18 |
| CMIP5 SST on AMIP5 net flux (R2) | 0.84 | 0.10 ± 0.02 $KW^{-1}m^2$ | 1.4E-5 | 18 |
| CMIP5-AMIP5 net flux on AMIP5 net flux (R3) | −0.84 | −0.81 ± 0.13 | 1.5E-5 | 18 |
| CMIP5 SST on CMIP5 net flux (R4) | 0.35 | – | 1.6E-1 | 18 |
| **Heat flux component and SST biases results** | | | | |
| CMIP5 SST bias on AMIP5 SW | 0.73 | 0.06$KW^{-1}m^2$ | 6.2E-4 | 18 |
| CMIP5 SST bias on AMIP5 LW | −0.44 | – | 6.5E-2 | 18 |
| CMIP5 SST bias on AMIP5 turbulent | 0.27 | – | 2.8E-1 | 18 |
| AMIP5 net on AMIP5 SW | 0.91 | 0.60 | 2.2E-7 | 18 |
| AMIP5 net on AMIP5 LW | −0.61 | −0.67 | 7.1E-3 | 18 |
| AMIP5 net on AMIP5 SW + LW | 0.87 | 0.91 | 2.3E-6 | 18 |
| AMIP5 net on AMIP5 turbulent | 0.35 | 0.70 | 1.6E-1 | 18 |
| AMIP5 SW on AMIP5 LW | −0.81 | −1.37 | 2.8E-8 | 18 |
| CMIP5-AMIP5 turbulent on CMIP5 SST | −0.73 | −4.8 $Wm^{-2}K^{-1}$ | 5.2E-4 | 18 |
| CMIP5-AMIP5 LW on AMIP5 SST | −0.63 | −1.3 $Wm^{-2}K^{-1}$ | 5.3E-3 | 18 |
| CMIP5 SW on AMIP5 SW | 0.96 | 1.0 | 2.0E-10 | 18 |
| CMIP5 LW on AMIP5 LW | 0.96 | 1.0 | 1.1E-10 | 18 |
| CMIP5 turbulent on AMIP5 turbulent | 0.30 | – | 2.3E-1 | 18 |
| CMIP5 net on AMIP5 net | 0.33 | 0.18 | 1.8E-1 | 18 |
| CMIP5 SST on CMIP5 SW | 0.75 | – | 3.6E-4 | 18 |
| AMIP5 net on CMIP5 SW | 0.86 | 0.54 | 4.0E-6 | 18 |
| **ZWML results** | | | | |
| CMIP5 ZWML bias on AMIP5 net flux | −0.72 | −0.18°$W^{-1}m^2$ | 6.7E-4 | 18 |
| AMIP5 ZWML bias on AMIP5 net flux | −0.44 | – | 1.0E-1 | 15 |
| CMIP5 ZWML bias on CMIP5 SST | −0.85 | – | 5.0E-5 | 15 |
| **Ocean heat content results** | | | | |
| CMIP5 300 m heat content on AMIP5 net flux | 0.68 | – | 1.8E-3 | 18 |
| CMIP5 1000 m heat content on AMIP5 net flux | 0.60 | – | 8.3E-3 | 18 |
| **AMOC strength result** | | | | |
| SST on AMOC maximum strength | −0.07 | – | – | 13 |

All fluxes are net downward surface fluxes. See Supplementary Table 1 for the individual models in each model set
AMOC is Atlantic meridional overturning circulation, SW is short-wave radiation, LW is long-wave radiation, turbulent is total turbulent flux, ZWML is zonal wind maximum latitude

derive and apply novel equilibrium mixed layer heat budget theory over the next three subsections. In the first subsection, by making a series of assumptions, we derive two simple analytical models of differing complexity (a consistent but more complete theoretical model with fewer assumptions is discussed in several extended theory subsections in the Methods). In the next subsection the expectations from these analytical models are compared with statistical analyses of the AMIP5/CMIP5 ensemble. In a third subsection, to link the AMIP5 net flux and CMIP5 SST biases to cloud process-representation deficiencies in the AMIP5 stand-alone atmospheric models we analyse their correlations on AMIP5/CMIP5 individual heat flux component biases.

**Equilibrium mixed layer heat budget theory**. In observations, SST is closely linked with ocean mixed layer temperature, $T_{OBS}$, which is controlled by the balance between the surface heat flux, $F_{OBS}$, and the combined horizontal and vertical ocean heat transport convergence into the mixed layer, $C_{OBS}$ (Fig. 2a)[33]. In any coupled model experiment, initialised from observational estimates, process representation deficiencies in the atmosphere and ocean model components cause errors in this heat budget. This results in evolving biases in the simulated mixed layer temperature, $T$, surface heat flux, $F$, and combined horizontal and vertical ocean heat transport convergence into the mixed layer, $C$.

At any location, approximate equilibration of the mixed layer with the atmosphere and upper-ocean typically occurs in coupled simulations within a few decades[34,35]. Close to observed or simulated equilibrium, the time tendency in mixed layer heat

capacity is small. To conserve heat in the mixed layer the observed and simulated sum of the heat flux terms must therefore also be small, i.e. $F + C \approx 0$ and $F_{OBS} + C_{OBS \approx} \approx 0$. Subtracting these two equations, and rearranging, we find that in a coupled model the net downward surface heat flux bias ($\Delta F = F - F_{OBS}$) must approximately cancel with the combined vertical and horizontal ocean heat transport convergence bias ($\Delta C = C - C_{OBS}$) so:

$$\Delta F \approx -\Delta C \qquad (1a)$$

We assume that the mixed layer temperature bias ($\Delta T = T - T_{OBS}$) is approximately equal to the SST bias. We can estimate $\Delta T$ and $\Delta F$ for any CMIP5 coupled model. In the Methods, we show that the equilibrium assumption is valid for our CMIP5 historical experiment multi-annual mean bias estimates.

We can decompose the coupled $F$ bias ($\Delta F$) into contributions from the stand-alone atmosphere model $F$ bias under realistic surface forcing ($\Delta F_A$) and a coupled $F$ response term ($\Delta F_R$), i.e. $\Delta F = \Delta F_A + \Delta F_R$ (Fig. 2b). We can estimate $\Delta F_A$ for any AMIP5 model from its estimated surface flux bias (with the robust assumption that errors in the AMIP5 prescribed surface boundary forcing, including SSTs, are small[24]). We define $\Delta F_R$, which is comprised of all of the coupled feedbacks, as the CMIP5 minus AMIP5 net flux bias difference ($\Delta F - \Delta F_A$). Our decomposition therefore becomes $\Delta F = \Delta F_A + (\Delta F - \Delta F_A)$, which is clearly valid as the positive and negative $\Delta F_A$ terms cancel.

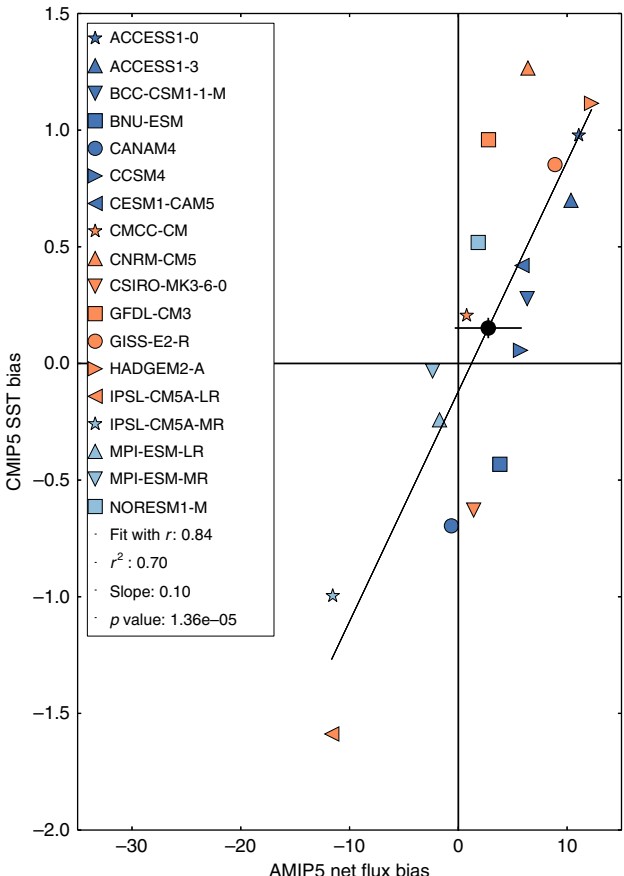

**Fig. 1** Linear regression of CMIP5 SST biases on AMIP5 net flux biases averaged over 40–60°S. These analyses were undertaken using the 18 models that provided suitable diagnostics for both CMIP5 and consistent AMIP5 experiments (see Methods and Supplementary Table 1). Our observational product uncertainty estimates are ~ 3 Wm$^{-2}$ for net flux and ~ 0.04 K for SST. Observational product errors do not affect the regression slope, correlation or $p$ values. However, the position of points for individual models and regression intercepts do include a contribution from observational error. The multi-model mean values are plotted with a solid black circle with a cross indicating their estimated observational uncertainties. The small multi-model mean bias estimates compared to the variations in the biases suggest that common structural errors in these parameters are not dominant for CMIP5 SST and AMIP5 net flux biases in this region and set of models

Similarly, we can decompose $\Delta C$ into a hypothetical stand-alone ocean model $C$ bias under realistic surface forcing ($\Delta C_O$) and a coupled $C$ response ($\Delta C_R$), i.e. $\Delta C = \Delta C_O + \Delta C_R$, where $\Delta C_R$ is defined as $\Delta C - \Delta C_O$. Unfortunately, for CMIP5, we cannot investigate if it is possible to provide first order estimates of $\Delta C_O$ from suitable stand-alone ocean model experiments because there are insufficient Ocean Model Inter-comparison Project (OMIP5[24]) ocean-only experiments with ocean models consistent with those employed in CMIP5. Furthermore, there are substantial known errors in the OMIP5 surface forcing sets and surface boundary conditions. Hence, although we could estimate $\Delta C$ from $-\Delta F$, we could not make use of it since $\Delta C_O$ is needed to estimate $\Delta C_R$.

$\Delta F_R$ and $\Delta C_R$ combine the local impact of all local and remote adjustments or feedbacks to the stand-alone model local F or C biases, which occur when all of the component models are coupled. $\Delta C_R$ includes the coupled impacts on C of errors in stand-alone atmospheric model momentum and freshwater forcing.

Combining these decompositions with Eq. 1a gives:

$$\Delta F_A + \Delta C_O \approx -(\Delta F_R + \Delta C_R) \qquad (1b)$$

For a single AMIP5/CMIP5 model, we cannot solve the decomposed equilibrium heat budget in Eq. 1b since we cannot estimate $\Delta C_O$ and $\Delta C_R$. However, we can make plausible simplifying assumptions to develop two analytical models of differing complexity, which predict regression relationships between the known $\Delta T$, $\Delta F$, $\Delta F_A$ and $\Delta F_R$ terms across the AMIP5/CMIP5 ensemble that we can analyse—good agreement would suggest our assumptions are valid, and vice-versa.

We assume a dominant negative linear dependence of $\Delta F_R$ on local $\Delta T$, with a negative sensitivity constant $\lambda_F$, i.e. $\Delta F_R \approx \lambda_F \Delta T$, as expected from bulk formulae[36] and observations[37]. For example, unrealistically high initial F (positive $\Delta F_A$) into the ocean will initiate warming (positive $\Delta T$), which will cause compensating increases in the latent, sensible and long-wave radiation heat fluxes out of the ocean surface (negative $\Delta F_R$), reducing the equilibrated F bias (smaller $\Delta F$). The Stefan–Boltzmann Law of black body radiation predicts a sensitivity of 4.8 Wm$^{-2}$K$^{-1}$ for surface upward long-wave radiation (at the observed 40–60°S area-mean SST of ~280 K). However, since SST, heat fluxes and the near-surface atmosphere are tightly coupled, clouds and water vapour will re-emit a fraction of the additional emitted long-wave radiation back to the surface and the humidity and temperature of the marine near surface boundary layer will adjust. This will reduce the magnitude of the long-wave and turbulent flux SST-sensitivity constants compared to bulk formulae estimates assuming no atmospheric response.

Similarly, we assume a dominant negative linear dependence of the coupled ocean heat transport convergence bias response, $\Delta C_R$, on local $\Delta T$ i.e. $\Delta C_R \approx \lambda_C \Delta T$, with a negative sensitivity constant, $\lambda_C$. This assumption could be valid for the Southern Ocean region because many models exhibit large positive SST biases for 40–60°S adjacent to large negative SST biases further north[13,14]. Northward wind-driven Ekman currents might therefore transport heat away from this region with an approximately linear dependence on the local area-mean SST bias. In other regions, however, this assumption is not expected to be valid since $\Delta C_R$ depends on horizontal and vertical gradients in ocean heat transport, which should not generally be linearly related to local $\Delta T$. For example, ocean heat transports depend on horizontal and vertical temperature gradients, mixed layer depths and currents, which in turn depend on density gradients and surface wind stresses.

Both of these coupled response approximations assume that residual contributions to $\Delta F_R$ and $\Delta C_R$ that are not linearly related to local SST are not dominant (In the Methods we present a more complete but consistent analytical model that includes these terms).

Combining these assumptions with Eq. 1b gives:

$$\Delta F_A + \Delta C_O \approx -(\lambda_F + \lambda_C) * \Delta T \qquad (2)$$

Re-arranging Eq. 2 gives our general case analytical model (Fig. 1b):

$$\Delta T \approx -(\Delta F_A + \Delta C_O)/(\lambda_F + \lambda_C) \qquad (3)$$

In this general case model, the SST bias in any coupled model at any location depends on the sum of the local stand-alone atmospheric model component F biases and ocean model

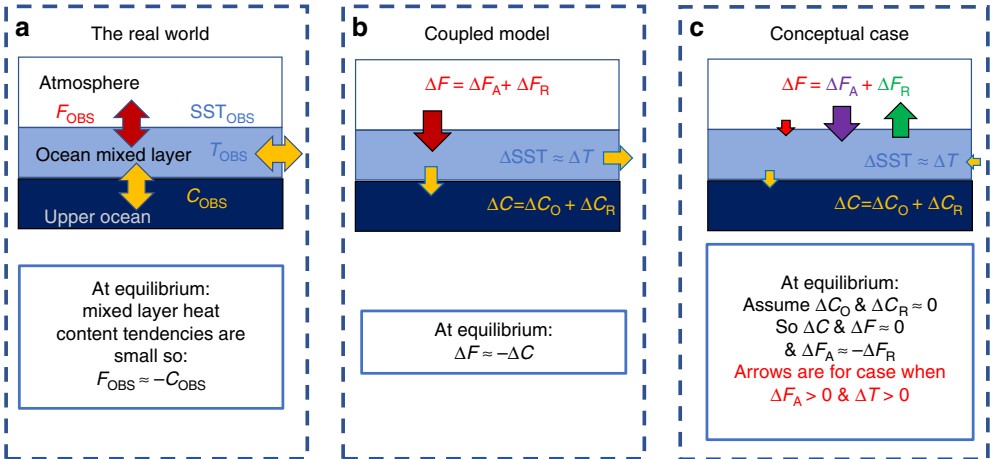

**Fig. 2** Schematic diagrams of the mixed layer heat budget. **a** the real world, **b** any coupled model, **c** a simplified coupled model conceptual case, where we assume a small stand-alone ocean model combined horizontal and vertical heat transport convergence bias ($\Delta C_O$) and a small coupled ocean heat transport convergence response ($\Delta C_R$). $F_{OBS}$ is the observed surface heat flux, $C_{OBS}$ is the observed combined vertical and horizontal ocean heat transport convergence and $T_{OBS}$ is the mixed layer temperature or SST. The simulated mixed layer temperature bias, $\Delta T$, is assumed to be equal to the SST bias. The simulated surface heat flux bias, $\Delta F$, can be decomposed into a stand-alone atmospheric model bias ($\Delta F_A$) and a coupled response ($\Delta F_R$). The simulated combined vertical and horizontal ocean heat transport convergence bias, $\Delta C$, can be decomposed into a stand-alone ocean model bias ($\Delta C_O$) and a coupled response ($\Delta C_R$). In (**c**) the simplified conceptual case since $\Delta C_O$ and $\Delta C_R$ are assumed small, $\Delta C$ & $\Delta F$ must both also be small. Hence, $\Delta F_A$ must be approximately compensated for by $\Delta F_R$

component C biases divided by the sum of the sensitivities to local SST of the coupled F and C responses.

We can also derive a much simpler conceptual case analytical model (Fig. 2c) by assuming that the stand-alone ocean model errors ($\Delta C_O$) and coupled C responses ($\Delta C_R$ and $\lambda_C$) are small compared to be the other terms in Eqs. 2 and 3:

$$\Delta F_A \approx -\Delta F_R \approx -\lambda_F \Delta T, \lambda_F < 0 \quad (4)$$

And:

$$\Delta T \approx -\Delta F_A/\lambda_F \approx \Delta F_R/\lambda_F, \lambda_F < 0 \quad (5)$$

This conceptual case model is useful since it provides simple relationships which only involve the terms we know. In this model, stand-alone atmospheric model errors must be balanced by an opposing surface flux response so the coupled flux biases are small. We will discuss the validity of the assumptions in this model compared to those in the general case model when we compare their expectations with the AMIP5/CMIP5 ensemble regression results (in the Methods we also consider three alternative simple conceptual models in which different pairs of terms on the right hand side of Eq. 3 are assumed small).

Our general and conceptual case analytical models provide expressions for $\Delta T$ given by Eqs. 3 and 5, respectively. These expressions would be expected to result in several relationships in regression analyses between variations across the AMIP5/CMIP5 ensemble in $\Delta T$, $\Delta F_A$, $\Delta F_R$ and $\Delta F$ assuming that $\lambda_F$ and $\lambda_C$ are consistent across the models. Since we cannot estimate $\Delta C_O$ and $\Delta C_R$ our general case model in Eq. 3 requires the additional assumption that $\Delta C_O$ and $\Delta F_A$ are approximately uncorrelated. We expect this assumption to be valid since $\Delta C_O$ and $\Delta F_A$ are biases in completely different stand-alone ocean and atmospheric model components, which are largely developed separately.

Correlations for these relationships will be limited by the regression residuals that include contributions from errors in all of the analytical model assumptions. For example, we expect $\lambda_F$ and $\lambda_C$ to vary at least to some degree between models. We also expect the $\Delta C_R$ and $\Delta F_R$ coupled responses to include a

component that does not depend linearly on $\Delta T$. $\Delta C_O$ variations will also contribute since they are assumed small in the conceptual case model and are unknown for the general case model. Strong correlations would suggest that these terms are small since correlations would be weak if any of these terms were dominant.

In the next subsection, we compare the AMIP5/CMIP5 ensemble regression results with the expectations from our two analytical models.

**Theoretical expectations and AMIP5/CMIP5 ensemble results.** From our simple conceptual case model in Eqs. 4 and 5, with negative $\lambda_F$, we expect and find several inter-related regression relationships across the AMIP5/CMIP5 ensemble for 40–60°S (Table 1). Since $\Delta F_R \approx \lambda_F \Delta T$ we expect a strong negative correlation of $\Delta F_R$ on $\Delta T$, i.e. CMIP5-AMIP5 net flux biases on SST biases ($r = -0.66$, $S = -5.5 \pm 1.6$ Wm$^{-2}$K;$^{-1}$ R1). Since $\Delta T \approx -\Delta F_A/\lambda_F$ we expect a strong positive correlation of $\Delta T$ on $\Delta F_A$, i.e. CMIP5 SST biases on AMIP5 net flux biases ($r = 0.84$ $S = 0.10 \pm 0.02$ KW$^{-1}$m$^2$ (Fig. 1; R2). Since $\Delta F_A \approx -\Delta F_R$ we expect a strong negative correlation of $\Delta F_R$ on $\Delta F_A$ i.e. CMIP5-AMIP5 net flux biases on AMIP5 net flux biases ($r = -0.84$; R3). Since $\Delta F = \Delta F_A + \Delta F_R \approx 0$ we expect a weak correlation of $\Delta T$ on $\Delta F$ i.e. CMIP5 SST biases on CMIP5 net flux biases ($r = 0.35$; R4).

The strong anti-correlation for $\Delta F_R$ on $\Delta T$ of $-0.66$ (R1) suggests validity in our assumption in both analytical models that the net downward coupled surface flux responses have a negative linear dependence on SST biases. Since $\Delta F_R \approx \lambda_F \Delta T$ the $\Delta F_R$ on $\Delta T$ regression slope provides a multi-model mean estimate of $\lambda_F$ of approximately $-5.5 \pm 1.6$ Wm$^{-2}$K$^{-1}$.

The agreement between the expectation from the simple conceptual case model and the R1 to R4 relationship results across the AMIP5/CMIP5 ensemble, including a strong correlation for $\Delta T$ on $\Delta F_A$, suggest that $\Delta C_O$ and $\Delta C_R$ cannot be dominant terms in the CMIP5 ensemble equilibrium mixed layer budgets (this inference is supported by the inconsistency between the results from the AMIP5/CMIP5 ensemble and expectations from the three alternative simplified conceptual case models presented in the Methods).

Our AMIP5/CMIP5 ensemble results also allow us to demonstrate some apparent limitations in the simplified conceptual case model, which suggest that some of its assumptions may not be completely valid. For example, from the conceptual case model, since $\Delta F_A \sim -\Delta F_R$, we would expect complete cancellation between $\Delta F_R$ and $\Delta F_A$ variations resulting in near-zero coupled flux bias variations and a very weak $\Delta T$ on $\Delta F$ correlation. Across the ensemble, the $\Delta T$ on $\Delta F$ correlation of 0.35 (R4) is weak. However, since $\Delta F_R$ and $\Delta F_A$ do not fully cancel, variations in $\Delta F$ are small but not negligible (the standard deviation of $\Delta F$ and $\Delta F_A$ are 3.7 $Wm^{-2}$ and 6.6 $Wm^{-2}$, respectively). Furthermore, the inverse of the slope of $\Delta T$ on $\Delta F_A$ regression (R2) provides a second estimate of $\lambda_F \sim -10.0$ ($-8.3$ to $-12.5$) $Wm^{-2}K^{-1}$, which appears to differ from the estimate derived from $\Delta F_R$ on $\Delta T$ (R1) of $-5.5$ ($-3.9$ to $-7.1$) $Wm^{-2}K^{-1}$.

In the general case model, the SST bias in any coupled model at any location depends on the sum of the local atmospheric model component F biases and ocean model component C biases, divided by the sum of the sensitivities of the coupled F and C responses to local SST (Eq. 3). The strong $\Delta T$ on $\Delta F_A$ correlation of 0.84 (R2) and supporting theory, strongly suggest that for 40–60°S variations across the AMIP5 model net flux biases are the dominant control on the variations in CMIP5 model SST biases. Seventy percent ($r^2 = 0.70$) of the CMIP5 $\Delta T$ variance is found to be explained by AMIP5 $\Delta F_A$. If, as we expect, $\Delta F_A$ and $\Delta C_O$ are independent then stand-alone ocean-ice model bias variations ($\Delta C_O$) cannot explain more than ~30% of $\Delta T$ variance (for details see Methods).

The general case model is in better agreement with the AMIP5/CMIP5 ensemble results than the simplified conceptual case model since it does not predict $\Delta F_A$ and $\Delta F_R$ to cancel completely (e.g. Eq. 2). The inverse of the $\Delta T$ on $\Delta F$ regression slope (R2) is given by $-(\lambda_C + \lambda_F)$. In combination with the estimate of $\lambda_F$ from $\Delta F_R$ on $\Delta T$ (R1), this suggests that the multi-model mean ocean heat transport convergence response sensitivity constant, $\lambda_C$, is approximately $-4.5$ ($-1.2$ to $-8.6$) $Wm^{-2}K^{-1}$. The uncertainties in this $\lambda_C$ estimate from the combined use of the R1 and R2 results are large but unavoidable since we cannot estimate $\Delta C_R$. They do not rule out the small $\lambda_C$ assumption used for the conceptual case model. However, the central estimate of $\lambda_C$ suggests that for the Southern Ocean 40–60°S region the theory benefits from including $\lambda_C$. If it becomes possible in future to estimate $\Delta C_R$ from OMIP/CMIP experiments then the $\Delta C_R \approx \lambda_C \Delta T$ relationship could provide a better-constrained estimate of $\lambda_C$.

The apparently small (<~30%) fraction of CMIP5 SST bias variance explained by ocean model C bias variance is a surprising result, particularly given that we find that in the HadGEM3 models (see Methods) SST biases are sensitive to ocean model resolution and parameters. For example, the change in SST bias from HadGEM3-GC2 to HadGEM3-GC3.1 appears to include a contribution from ocean model changes (Supplementary Figure 2). Note, however, that this small fraction is for this set of models, and includes the impact of any enhancement of component model F and C bias variances by outlying models (see Methods for details) and any under-sampling of component model F and C bias variances due to common-structural errors (the $\lambda_F$ and $\lambda_C$ estimates could also be sensitive to the choice of models). Small multi-model mean AMIP5 net flux and CMIP5 SST biases (Fig. 1) suggest that under-sampling of component model errors is not dominant for this region and set of models, despite known structural errors, e.g. underestimated cloud brightness[15], low-resolution ocean models or too shallow summer ocean mixed layers[38,39]. Conceivably, cross-correlations of both $\Delta T$ and $\Delta F_A$ with other local or remote parameters could also

influence this result. However, this would require that CMIP5 coupled SST bias variations also be driven by an important mechanism which happened to also be correlated with AMIP5 stand-alone atmospheric model F biases which we do not expect or find any evidence of (see Methods).

Even given these caveats, the high fraction of SST bias variance explained by AMIP5 surface flux biases clearly contrasts with published singular value decomposition analyses[40], which suggest that inter-model Southern Ocean SST bias variations are related to variations in the strength of the Atlantic Meridional Overturning Circulation (AMOC). We also find an extremely weak correlation of $-0.07$ between Southern Ocean 40–60°S area-mean SST and AMOC strength across 13 coupled CMIP5 models.

**The role of heat flux component biases in causing SST biases.** To link AMIP5 net flux and associated CMIP5 SST biases to deficiencies in AMIP5 atmospheric model characteristics we also consider regression analyses of individual heat flux component area-mean biases for 40–60°S. AMIP5 surface net flux biases depend on their biases in individual flux components, which in turn depend on their biases in other atmospheric column parameters. For example, AMIP5 net surface downward short-wave biases have been shown to be strongly linked to biases in cloud characteristics, e.g. particularly deficiencies in simulated cloud amount and brightness[15]. By contrast, the AMIP5 net downward long-wave biases are expected to be strongly influenced by simulated cloud base height, cloud base temperature, and moisture errors. AMIP5 sensible and latent heat flux biases are expected to depend on the representation of numerous atmospheric parameters, including boundary layer characteristics.

AMIP5 net flux biases are strongly correlated with AMIP5 short-wave and total radiative flux biases ($r = 0.91$ and $r = 0.87$, respectively, Table 1). AMIP5 net flux biases are anti-correlated with their long-wave biases ($r = -0.61$) because long-wave biases are anti-correlated with the mostly larger short-wave biases ($r = -0.81$; Table 1). AMIP5 net flux biases are weakly correlated with turbulent flux biases ($r = 0.35$). Short-wave bias variations across the models are therefore the dominant driver of variations in AMIP5 net flux biases, mainly because short-wave inter-model variations are largest (Table 2; Supplementary Figure 3). As expected, the correlation of CMIP5 SST biases on AMIP5 short-wave flux biases is also much stronger ($r = 0.73$) than those on AMIP5 long-wave flux biases and AMIP5 total turbulent flux biases ($r = -0.44$ and $r = 0.27$, respectively). Hence, local AMIP5 short wave bias variations linked to cloud representation errors are the dominant driver of the AMIP5 net flux bias variations, which in turn appear to be the main driver CMIP5 SST bias variations.

The estimated surface net heat flux coupled response sensitivity constant, $\lambda_F$, of 5.5 $Wm^{-2}K^{-1}$ primarily arises through contributions from total turbulent and long-wave flux responses (short-wave response sensitivities are much weaker). The CMIP5-AMIP5 flux component bias on SST bias regression sensitivity constants are $-4.8$ and $-1.3$ $Wm^{-2}K^{-1}$ for the net downwards total turbulent and long-wave flux, respectively (Table 1). The sensitivity of $-1.3$ $Wm^{-2}K^{-1}$ for net downward long-wave suggests that around three quarters of the emitted upward long-wave estimated from Stefan's law (~ 4.8 $Wm^{-2}K^{-1}$) is re-emitted back towards the surface. The total turbulent flux sensitivity of $-4.8$ $Wm^{-2}K^{-1}$ is broadly consistent with observational estimates of its sensitivity for the Southern Ocean region derived from ERAI[35] of < 10 $Wm^{-2}K^{-1}$. These estimates are both of considerably smaller magnitude than estimates of more than 20 $Wm^{-2}K^{-1}$ for the total turbulent flux sensitivity from bulk formulae for this region ignoring the atmospheric response.

**Table 2 Stand-alone atmospheric model surface heat flux component biases and Total Absolute Flux Biases and coupled model SST biases averaged over 40–60°S**

| 40–60°S area-mean values | Atmos. only estimated short-wave flux bias ($Wm^{-2}$) | Atmos. only estimated long-wave flux bias ($Wm^{-2}$) | Atmos. only estimated turbulent flux bias ($Wm^{-2}$) | Atmos. only estimated net flux bias ($Wm^{-2}$) | Atmos. only estimated TAFB ($Wm^{-2}$) | Coupled estimated SST bias (K) |
|---|---|---|---|---|---|---|
| Observational uncertainty | ±1.0 | ±6.0 | ±7.0 | ±3.0 | ±8.0 | ±0.04 |
| AMIP5 or CMIP5 mean | 1.9 | −5.7 | 6.5 | 2.8 | 21.3 (14.1) | 0.15 |
| AMIP5 or CMIP5 STD | 10.0 | 5.9 | 3.2 | 6.6 | 9.6 | 0.77 |
| HadCM3 | 0.0 | −13.1 | 15.6 | 2.5 | 28.7 | 0.38 |
| HadGEM1 | 1.4 | −3.7 | 7.1 | 4.8 | 11.9 | −0.27 |
| HadGEM2 | 9.4 | −6.3 | 9.1 | 12.2 | 24.8 | 1.20 |
| HadGEM3-GC2 | 7.8 | −1.9 | 10.0 | 15.6 | 19.7 | 2.59 |
| HadGEM3-GC3.1 | 2.0 | 0.6[a] | 3.0[a] | 5.5 | 5.6[a] | 0.57 |
| GC2 to GC3.1 change | Apparent improvement | Both within uncertainty | Apparent improvement | Apparent improvement | Apparent improvement | Apparent improvement |

The Hadley Centre coupled climate models are presented together with the multi-model means and standard deviations (STD) for the 18 AMIP5/CMIP5 models. For the AMIP5/CMIP5 models the multi-model mean of the individual model total absolute flux biases (TAFB) estimates is presented but the TAFB estimated from the multi-model mean component biases is also included in brackets. Area-mean changes between HadGEM3-GC2 and HadGEM3-GC3.1 are classified as either apparent improvements (given our observational uncertainties) or both within uncertainties (when models are both within the observational uncertainties so changes should be considered as differences rather than improvements)
[a]s indicate parameters for which HadGEM3-GC3.1 biases are within the estimated observational uncertainty

In stand-alone ocean-only model experiments, the surface flux responses estimated from bulk formulae cannot be damped by atmospheric adjustments. We therefore expect $\lambda_F$ to be considerably more negative. This would result in much smaller SST biases than those in coupled models even if atmospheric forcing errors were similar. Furthermore, we expect the overly strong ocean-only evaporative response to temperature (without the counteracting known precipitation SST response) to result in a substantially overestimated upwards freshwater flux response to SST biases. These unrealistic feedbacks could at least in part explain the need for sea surface salinity restoring towards observational estimates in ocean-only models.

CMIP5 on AMIP5 correlations are strong for both the short-wave ($r = 0.96$) and long-wave ($r = 0.96$) flux components. These high correlations suggest that their coupled SST responses are small compared to their AMIP5 inter-model variations. In consequence, 40–60°S CMIP5 short wave biases are strongly cross-correlated with AMIP5 net flux biases ($r = 0.86$), which our results suggest is the dominant driver of SST biases. This explains the previously identified[17] correlation of SST biases on coupled short wave flux biases (we find $r = 0.75$). The CMIP5 on AMIP5 correlation is very weak for both total turbulent ($r = 0.30$) and net fluxes ($r = 0.33$), mainly due to their smaller AMIP5 inter-model variations and strong coupled responses to local coupled SST biases.

**Impacts of AMIP5 net flux and CMIP5 SST biases.** We first investigate coupled wind feedbacks using regression analyses of biases in AMIP5/CMIP5 near surface (10 m) zonal wind maximum latitude (ZWML) on biases in AMIP5 net flux and CMIP5 SST. ZWML biases are expected to influence ocean thermal structure well below the mixed layer via Ekman pumping caused by wind stress curl[12]. We also expect biases in atmospheric model net flux, and associated coupled ZWML biases, to adversely influence simulated heat (and carbon) uptake under climate forcing (since vertical heat transports depend on both stratification and Ekman pumping). In climate model projections, the position of the Antarctic Circumpolar Current core, which is in geostrophic balance with ocean thermal structure, is also often correlated with the latitude of the wind stress maximum over decadal timescales[12,41].

CMIP5 ZWML biases are anti-correlated ($r = −0.72$) with AMIP5 net heat flux biases (Fig. 3; Table 1). However, AMIP5 ZWML biases are only weakly anti-correlated with AMIP5 net flux biases ($r = −0.44$), indicating that the internal atmospheric dynamical link is much weaker. CMIP5 ZWML biases are more strongly anti-correlated with CMIP5 SST biases ($r = −0.85$). Hence, the dynamical link between CMIP5 ZWML biases and AMIP5 net flux biases must primarily arise through the impact of atmospheric net fluxes on evolving coupled SST, which in turn feeds back onto ZWML and SST. Previously, this link has been identified solely within CMIP5, which precluded clear attribution to atmospheric model net flux errors[17]. These results highlight the potential for AMIP5 net flux biases to influence Southern Ocean wind responses to climate forcing[42]. Few CMIP5 models appear to represent both ZWML and SST well (given their estimated observational uncertainties). Most models with more realistic smaller equatorward ZWML biases appear to have less realistic larger positive SST and downward net flux biases, and vice-versa (Fig. 3).

Next, we investigate the impact of AMIP5 net flux biases and associated SST-wind feedbacks on ocean thermal structure. The 0–300 m and 0–1000 m ocean heat content bias correlations on AMIP5 net flux biases are weaker than those for CMIP5 SST biases on AMIP5 net flux biases ($r = 0.68$, $r = 0.60$ and $r = 0.84$, respectively; Table 1; Supplementary Figure 4). The weaker correlation for heat content biases on AMIP5 net flux biases for deeper upper ocean layers could result from either increasing contributions from stand-alone ocean-model heat transport convergence biases or a weaker association of heat content biases with SST biases and their associated surface heat flux responses.

The impact of atmospheric model net flux biases, and associated wind feedbacks, on ocean thermal structure is also investigated by comparing CMIP5 temperature composites over models exhibiting high (HIF, 7 models) and near zero or low (NZF, 7 models) AMIP5 downward net flux biases. HIF is significantly warmer than NZF in the upper 1500 m north of ~ 58°S (Fig. 4). There is also an apparent north-south dipole in thermal structure difference extending to 3000 m; with cooler

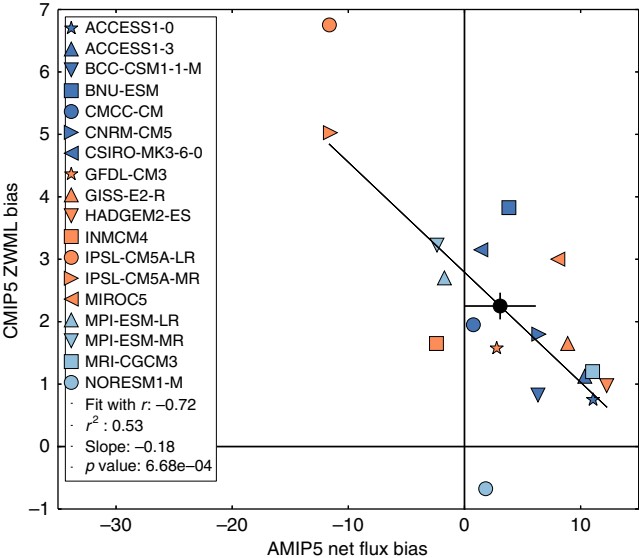

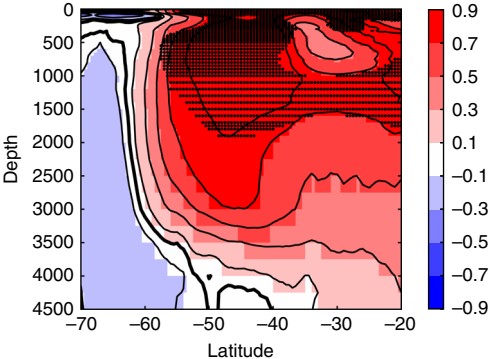

**Fig. 4** The difference between zonal mean CMIP5 historical experiment ocean temperature composites by high and near zero or low estimated AMIP5 net flux biases. Stippling is used to indicate where differences are statistically significant (see Methods)

**Fig. 3** Linear regression of estimated biases in the latitude of the CMIP5 maximum zonal mean 10 m westerly wind on the estimated 40–60°S AMIP5 net flux biases. Zonal wind maximum latitude (ZWML) biases are relative to ERA Interim with positive values indicating northward biases. Our observational product uncertainty estimates are ~ 3 W m$^{-2}$ for net flux bias and ~ 0.2° for ZWML (see Methods). Observational product errors do not affect the regression slope, correlation or $p$ values. However, the position of points for individual models and regression intercepts do include a contribution from observational error. The multi-model mean values are labelled with a solid black circle with a cross through it representing their estimated observational uncertainties. The large multi-model mean CMIP5 ZWML bias but small multi-model mean AMIP5 net flux bias suggest that other common structural errors influence errors in CMIP5 ZWML for this set of models

waters to the south of ~ 58°S and warmer waters to the north for HIF (these models tend to have higher 40–60°S SST and more poleward ZWML). Differences below 1500 m are not statistically significant (possibly due to relatively small composite sizes). This deep dipole pattern is, however, broadly consistent with the observed change in thermal structure linked to the poleward wind migration associated with the Southern Annular Mode increase over recent decades[11,43].

**Stand-alone atmospheric model heat flux component biases.** Across AMIP5, considerable inter-model variations are evident in all individual heat flux component biases (Table 2; Supplementary Figure 5). We term the sum of the absolute value of area-mean short-wave, long-wave and turbulent flux biases for each model the 'Total Absolute Flux Bias', TAFB. (i.e. TAFB = |short-wave bias| + |long-wave bias| + |turbulent flux bias|). The simulated response of net surface flux to climate forcing arises from the sum of the individual heat flux component responses, which depends on the fidelity of their present day magnitudes. TAFB provides a first order metric of the overall fidelity of their present day simulation. Achieving a small net flux bias with a small TAFB would therefore increase confidence in a model's coupled net flux response (i.e. 'right for the right reasons').

Many models with small net flux biases do not appear to have small estimated TAFB (Fig. 5), given our observational uncertainty estimates of ~ 3 and ~ 8 W m$^{-2}$ for net flux and TAFB, respectively (see Methods). This suggests that there may be considerable error cancellation between heat flux component biases. Cancellation between short-wave and long-wave biases is evident ($r = -0.81$, Table 1), which could be expected to result

from errors in cloud amount, thickness and/or brightness caused by deficiencies in parameterised cloud microphysics, particularly for mixed phase cloud. For coupled models, it is to some degree possible to adjust regional SST or net TOA flux by tuning parameter values, particularly in cloud microphysics[44,45]. If tuning were undertaken without regard for errors in the atmosphere-only surface heat flux components then it would be likely to introduce error cancellation[46], analogous to an implicit flux correction. This could affect the fidelity of the simulated response to climate forcing[47].

**A route to improve the models.** Over a succession of Hadley Centre climate models (HadCM3-A, HadGEM1-A, HadGEM2-A, HadGEM3-GC2-A to HadGEM3-GC3.1-A; see Methods), involving over 20 years of model development, estimated stand-alone atmospheric model TAFB have generally decreased for more recent configurations (Fig. 5; Table 2). An exception is HadGEM1 that had small long-wave biases due to cancellation between cloud height and cloud amount, highlighting that TAFB is not a comprehensive metric of model fidelity. However, from HadCM3-A to HadGEM3-GC2-A estimated net flux biases generally appear to have increased. In combination, we interpret these findings as evidence for the progressive removal of cancelling errors through improved process-representation (this is an example of how improving models can often initially make key biases worse). Subsequently, in-depth process evaluation and targeted development for HadGEM3-GC3.1-A over several years has considerably reduced both the estimated net flux bias and estimated TAFB simultaneously compared to HadGEM3-GC2.

In combination with improvements to ocean mixing, this has contributed to a ~70% reduction in the coupled SST biases for HadGEM3-GC3.1[48] compared to HadGEM3-GC2 (Fig. 6; Table 2 and Supplementary Figure 2). Crucially, new mixed phase cloud[49] and aerosol[50,51] schemes were implemented, improving the representation of clouds and radiation characteristics. This represents additional evidence to support our inference that Southern Ocean coupled SST biases are caused by local atmospheric model cloud-related surface flux biases. An ongoing effort to improve future configurations of HadGEM3 is focussed on improving the representation of cloud phase, particularly super-cooled liquid[52].

**Discussion**
Novel mixed layer heat budget theory with simplifying assumptions predicts that coupled model SST biases arise through the mixed layer equilibration process, which primarily involves

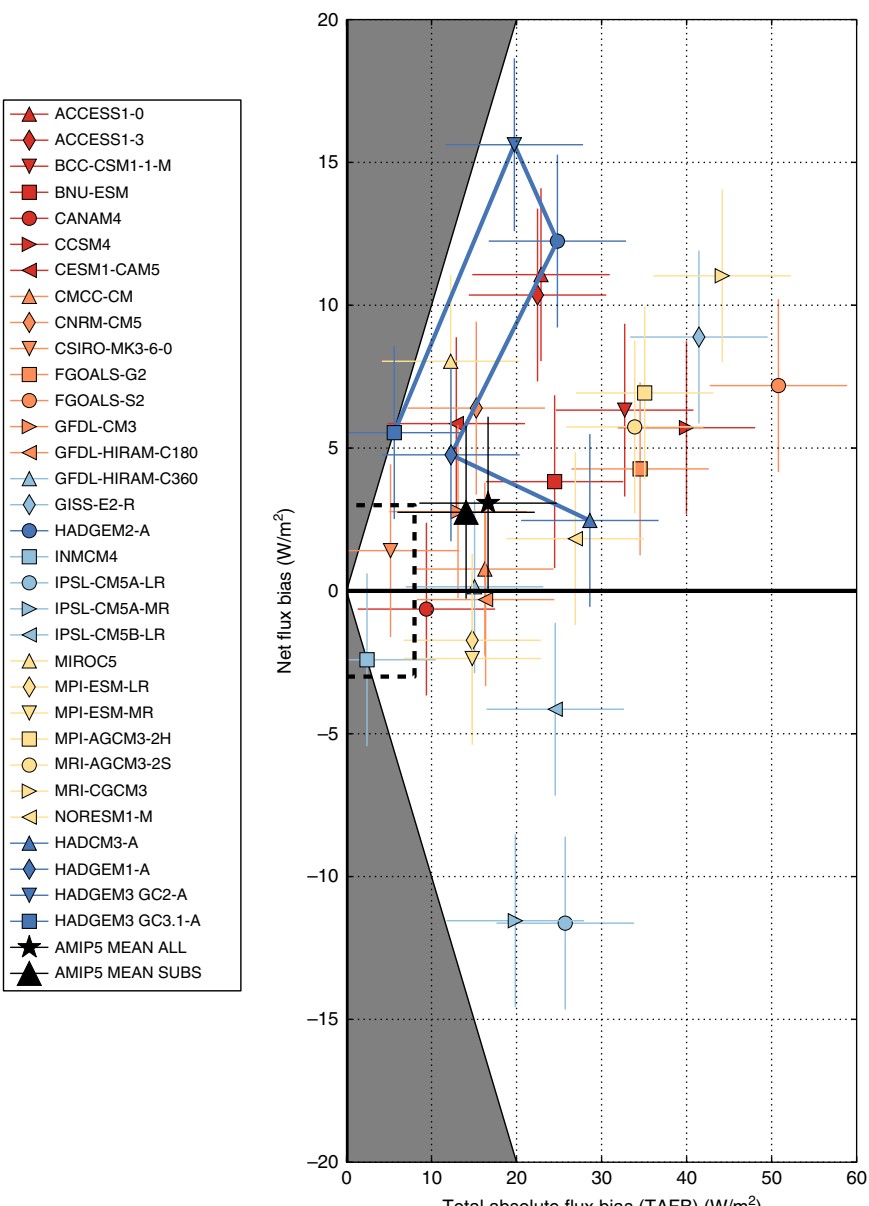

**Fig. 5** Relationship between estimated net flux biases and estimated total absolute flux biases for the AMIP5 and Hadley Centre atmospheric models. Multi-model means for all AMIP5 models (AMIP5 MEAN ALL) and for the subset of 18 AMIP5 models with CMIP5 SST (AMIP5 MEAN SUBS) are also included. The Hadley Centre models are labelled with dark blue symbols and connected in order of release date by a solid dark blue line. Note that these model bias estimates include a contribution from observational errors. Our indicative observational uncertainty estimates of 8 Wm$^{-2}$ for TAFB and 3 Wm$^{-2}$ for net flux (see Methods) are represented by crosses centred on each model symbol. A box enclosing all the model symbols within our estimated observational uncertainties is also marked with a dashed black line. The region where TAFB is less than net flux bias is shaded in grey since estimated TAFB must be equal to or larger than estimated net flux bias

coupled responses in local surface heat flux, *F*, and combined horizontal and vertical ocean heat transport convergence, *C*, which are linearly related to local SST biases. SST biases depend on the sum of the local stand-alone atmospheric model F and ocean model C biases, divided by the sum of the local F and C coupled SST response sensitivity constants. This means that models could have small SST biases by having too large C and F sensitivities to SST even if stand-alone component model errors were large (and vice-versa).

For the Southern Ocean 40–60°S region evidence from an interpretive framework combining the theory with regression analyses across this AMIP5/CMIP5 ensemble suggests that CMIP5 coupled climate model SST bias variations are primarily

caused by stand-alone atmospheric model net flux bias variations. These atmospheric model net flux bias variations are mainly associated with variations in short-wave radiation biases, which have previously been linked to cloud-representation deficiencies[15]. This provides strong new evidence for a causal link between clouds, radiation, SSTs, and atmospheric circulation over the Southern Ocean.

As expected from theory, variations in the AMIP5/CMIP5 coupled surface heat flux, *F*, response are linearly related to local temperature variations with a negative sensitivity constant. The sensitivity of the ocean C responses to SST biases remains uncertain but appears to also be important for this region. The surface flux responses to SST biases reduce the inter-model

variations in coupled model net flux biases compared to those in the stand-alone atmospheric models. However, erroneous supplies of heat and moisture to the coupled atmosphere persist through compensation of atmospheric model short-wave biases by coupled turbulent and long-wave flux feedbacks. This could contribute to causing known coupled model atmosphere biases in temperature, wind, heat transport, and Top-Of-Atmosphere (TOA) energy fluxes[18]. For example, AMIP5 net flux biases, and associated CMIP5 SST biases, appear to be linked with biases in the latitude of the CMIP5 zonal mean eastward wind maximum, together adversely influencing sub-surface ocean thermal structure.

The conclusions summarised above do not depend on the accuracy of our observational products. Aiming to limit

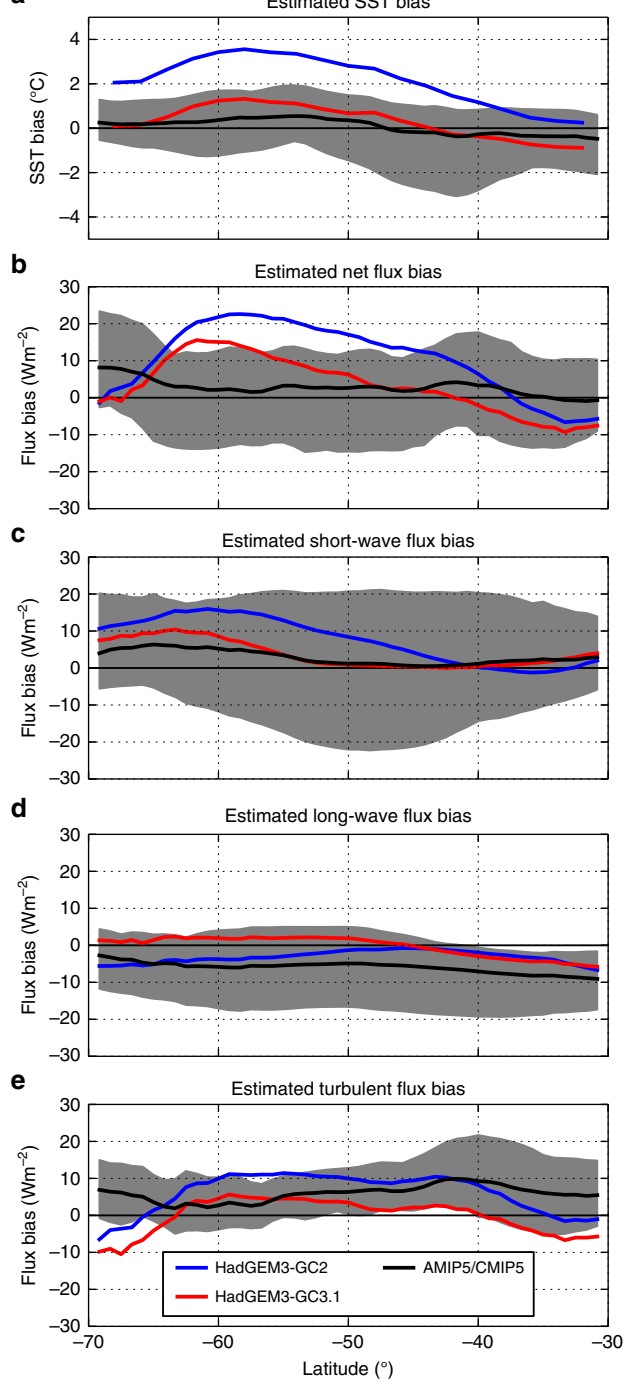

observational errors in our bias estimates for individual atmospheric AMIP5 models we employ a novel method to estimate observational heat flux components. Most AMIP5 models with small net flux biases appear to achieve this through considerable error cancellation between heat flux components. Improved cloud process-representation over successive Hadley Centre models has demonstrated an apparent route to improve the representation of heat flux components in stand-alone atmospheric models and reduce error cancellation. This has considerably improved SST biases in HadGEM3-GC3.1 the Hadley Centre CMIP6 model. Our estimates of uncertainties in our observational products, although unavoidably basic, are generally considerably smaller than the substantial model biases, providing confidence in these inferences.

The results shed light on one of the most long-standing and pervasive atmospheric and climate model deficiencies. They demonstrate a pathway to improve the next generation of coupled climate models and their projections in this important[53] and vulnerable[54] region for global climate. Effort needs to be prioritised to improve the representation of Southern Ocean air-sea flux processes including cloud microphysics in atmospheric models[55] (this would also improve atmospheric re-analyses that suffer from similar biases[15]). This will require improvements to the accuracies of observational estimates of cloud characteristics, surface flux components and ocean temperature, and their uncertainties (particularly if the fidelity of the models improves reducing their currently substantial biases).

In future, this methodology could prove useful in interpreting the causes of coupled SST biases in other regions, particularly those where correlations between CMIP5 SST biases and AMIP5 net flux biases are strong (Supplementary Figure 4). Similar approaches could also be employed to interpret causes of other coupled model parameter biases. Ultimately, this could help to constrain uncertainties in climate projections by identifying physical relationships between observable present day biases and uncertain aspects of the simulated climate responses, i.e. provide emergent constraints.

## Methods
**Heat flux observations**. For all our observational heat flux component estimates, an averaging period of 2001–2007 inclusive was used to permit inter-comparison between available radiative and turbulent flux product climatologies over their consistent overlapping period. All surface heat fluxes presented are net downward fluxes.

We employ a novel method[27,56] to estimate globally-balanced net downward surface flux through the atmospheric lower boundary from observed Top-Of-

**Fig. 6** Simulated ocean-area zonal mean coupled SST biases and stand-alone atmospheric model heat flux component biases for HadGEM3-GC3.1 and HadGEM3-GC2. (**a**) SST biases, (**b**) net downwards surface heat flux biases, (**c**) net downwards surface short-wave radiation flux biases, (**d**) net downwards long-wave radiation flux biases and **e** total downwards turbulent heat flux biases for HadGEM3-GC2(-A) (blue) and HadGEM3-GC3.1(-A) (red). The AMIP5/CMIP5 multi-model mean (black line) and spread (grey shading) for the 18 consistent AMIP5/CMIP5 models (see Methods) are included on all panels. The HadGEM3-GC2 and HadGEM3-GC3.1 SST biases are for present day control runs for years 50 to 100. The maximum estimated zonal mean observational errors between 35 and 65°S for SST, net flux, short-wave flux, long-wave flux and turbulent flux are 0.3 K, 7 Wm$^{-2}$, 3 Wm$^{-2}$, 10 Wm$^{-2}$ and 14 Wm$^{-2}$, respectively (see Methods). At most latitudes, there appear to have been improvements between HadGEM3-GC2 and HadGEM3-GC3.1 for SST, net flux and short wave, given these estimated observational errors. However, the HadGEM3-GC2 to HadGEM3-GC3.1 differences for estimated long-wave and total turbulent flux biases should be interpreted as changes not improvements (see Table 2 for equivalent 40–60°S area-mean results)

Atmosphere (TOA) net radiative fluxes[28] minus the total mass-consistent energy divergence and storage tendencies from the ERA-Interim[29] reanalysis. This is based on a method by Trenberth and co-authors using atmospheric column energy budgets[30,57,58]. This method estimates total energy flux through the lower atmospheric boundary, which for oceanic regions is approximately equal to the heat flux into the ocean, for regions and averaging periods over which latent heat fluxes associated with changes in sea-ice volume are small.

This approach is adopted since there are considerable and poorly quantified uncertainties in the observational estimates of the heat flux components, due to the scarcity of observations over ocean regions of near-surface atmospheric humidity, air temperature and winds; and direct estimates of heat fluxes[25,26,59]. Errors accumulate in net flux estimates since relatively small net fluxes are comprised of large seasonally-reversing net fluxes, which are in turn estimated from large opposing radiative and turbulent heat flux components. As a result, observations often yield unrealistic large absolute global mean surface net flux values, when radiative and turbulent heat flux products are added together, hindering their use for the assessment of models[59].

Our method avoids error accumulation from individual heat flux components as it directly estimates net flux. TOA radiative fluxes are observed[28]. Assimilated observations constrain reanalysis column energy storage tendencies and divergences more than they constrain their surface heat flux components (substantial simulated TOA and surface short-wave flux biases tend to also cancel, at least partially, in the energy divergences). This is evidenced (Supplementary Figure 1) by the considerable reduction in the spread in zonal mean net surface fluxes estimated using our method ($<\sim10\,\mathrm{W\,m^{-2}}$) for several reanalyses[27] compared to the spread in their directly simulated zonal mean net downward surface fluxes ($>\sim40\,\mathrm{W\,m^{-2}}$). We use net downward surface short-wave and long-wave radiation flux component estimates from CERES-EBAF (version 2.7)[31]. We then estimate the total turbulent flux as a residual at each grid point by subtracting CERES-EBAF derived net surface downwards total radiative flux from our net flux estimate. As discussed, this approach is adopted since the spread in observational estimates of turbulent fluxes is considerably larger than the spread between observational estimates of radiative heat fluxes[26]. For example, for 40–60°S the OAFLUX[60] vs. SEAFLUX[61] total turbulent flux difference is $25\,\mathrm{W\,m^{-2}}$ and the CERES-EBAF vs. ISCCP[62] total radiative flux difference is $6\,\mathrm{W\,m^{-2}}$. To assess the fidelity of the flux component balance for each model we also estimate total absolute flux bias (TAFB), which is defined as the sum of the absolute value of the time and area-mean short-wave, long-wave and turbulent flux component estimated biases.

**Heat flux observational uncertainties**. Reliable quantitative estimates of observational heat flux component uncertainties are not available, nor are the associated spatial and temporal error correlations needed to estimate the uncertainty reduction associated with spatial and temporal averaging. We are therefore only able to provide basic indicative uncertainty estimates.

For 40–60°S for the period 2001–2007, we applied the energy divergence method to estimate net flux using different re-analyses, which employ different models, assimilation systems and assimilated observational data[27]. Area-averaged net flux differences relative to our reference estimate (derived from ERA-Interim[29]) are 3.1, 3.4, 1.4 and $2.1\,\mathrm{W\,m^{-2}}$ for the ERA20C[63], JRA55C[64], JRA55[65] and MERRA[66] reanalysis, respectively. This suggests 40–60°S area-mean uncertainties for our net flux observational estimate are $\sim\pm3\,\mathrm{W\,m^{-2}}$, which we assume is indicative of its standard error. Note, however, that this estimate does not include structural errors common to all of the re-analyses. Differencing area means for CERES-EBAF and ISCCP suggests observational uncertainties (indicative of standard errors) are approximately ±1, 6 and $6\,\mathrm{W\,m^{-2}}$ for net downward surface short-wave, long-wave and total radiative flux, respectively. Note that CERES-EBAF and ISCCP are the only two available gridded global estimates of surface radiative flux components derived from observationally constrained atmospheric column radiative transfer methods. Assuming that the observational errors in the net flux and total radiative terms are independent, and can therefore be added in quadrature, the uncertainty in turbulent flux will be $\sim7\,\mathrm{W\,m^{-2}}$.

The same method is employed to estimate uncertainty in our observational zonal mean heat flux at each directly simulated point on the model grid. The standard error in our zonal mean net flux observational product at all latitudes is assumed to be $7\,\mathrm{W\,m^{-2}}$, the maximum value between 35°S and 65°S of the standard deviation of net flux estimates derived from applying our method to several reanalyses products[27] (its mean value between these latitudes is $\sim4\,\mathrm{W\,m^{-2}}$). The estimated maximum standard errors between 35°S and 65°S in our zonal mean observational estimates of short-wave, long-wave, total radiative fluxes, and turbulent fluxes are 3, 10, 12 and $14\,\mathrm{W\,m^{-2}}$ (mean values are 1, 6, 6 and$10\,\mathrm{W\,m^{-2}}$, respectively).

Note that our area-mean radiative flux uncertainty estimates based solely on differences between CERES-EBAF and ISCCP are smaller than uncertainties quoted for CERES-EBAF[31] for downward long-wave of $9\,\mathrm{W\,m^{;-2}}$ for upward long-wave of $8\,\mathrm{W\,m^{;-2}}$ for downward short-wave of $6\,\mathrm{W\,m^{;-2}}$ and for upward shortwave of $3\,\mathrm{W\,m^{-2}}$. However, the quoted estimates are for a single annual mean rather than a 2001–2007 mean; are based on CERES prior to energy balance adjusted fluxes (EBAF) bias corrections; and apply to any 20° latitude area mean (not 40–60°S) so could potentially be influenced by larger uncertainties in other regions. Our longer 2001–2007 averaging period would reduce random errors compared to CERES

annual error estimates. If errors are correlated then error cancellation will occur when these components are combined (e.g. if cloud amount is over-estimated, net downward short-wave will be less positive and net downward long-wave will be less negative)[67].

For 40–60°S area-mean TAFB, the use of absolute bias estimates complicates estimation of the errors[67]. We therefore obtain an approximate estimate for the TAFB error by simulating 10,000 random samples of model flux component biases drawn from independent normal distributions with means and standard deviations given by those of our AMIP5 bias estimates (Table 2). We similarly assume observational errors are independent and normally distributed with standard deviations given by the above observational heat flux component standard error estimates and zero means i.e. we ignore any structural errors. This method suggests the random standard error in TAFB is $\sim8\,\mathrm{W\,m^{-2}}$. The modelled TAFB error distribution also suggests that our TAFB estimates may on average be biased low by $\sim3\,\mathrm{W\,m^{-2}}$. To maintain consistency between our heat flux component bias estimates and our TAFB estimates we do not apply this bias correction in Fig. 5 or Table 2. Since the same $3\,\mathrm{W\,m^{-2}}$ offset would be applied to all the models and uncertainty box in Fig. 5, correcting this bias would not in any way change our inferences. This $8\,\mathrm{W\,m^{-2}}$ uncertainty in the 40–60°S area-mean TAFB estimate is substantial and could be worsened by unquantifiable structural errors. Hence, our TAFB results should be interpreted with some caution. However, it should be noted that this uncertainty of $\sim8\,\mathrm{W\,m^{-2}}$ is considerably smaller than the spread in area-mean TAFB across the models of $\sim50\,\mathrm{W\,m^{-2}}$, suggesting that we should be able to robustly separate good from poor TAFB models.

A summary of the area-mean flux component observational uncertainties is presented in Table 2, together with the AMIP5 estimated bias means and standard deviations, and Hadley Centre model biases. Note that the same observational error will apply to all models so relative changes between models are independent of observational errors. For AMIP5 net flux, short-wave flux and TAFB, the observational uncertainties are generally considerably smaller than the mostly substantial model biases (Fig. 1; Fig. 5; Supplementary Figure 5). There appear to be improvements in 40–60°S area-mean fluxes between HadGEM3-GC2 and HadGEM3-GC3.1 for net flux, short wave, turbulent flux and TAFB given the estimated observational errors. For the zonal mean bias estimates in Fig. 6, at most latitudes there appear to have been improvements between HadGEM3-GC2 and HadGEM3-GC3.1 for net flux and short-wave and TAFB. Note, however, that turbulent and long-wave flux zonal mean bias estimates for both HadGEM3-GC2 and HadGEM3-GC3.1 are within the observational uncertainty, so these differences should be interpreted as changes rather than improvements.

The large uncertainties in observational estimates of air-sea flux components, and the lack of adequate information to more accurately quantify them, remain key barriers to progress on understanding model biases. There are considerable ongoing efforts to address this problem, including the Southern Ocean Observing System (SOOS) Air-Sea Flux working group.

**Regression and multi-model mean analyses**. For linear regression analyses we employ the terminology 'a regression of $Y$ on $X$' to refer to evaluating $Y$ as a linear function *of* $X$ plus errors. Since we generally quote correlation together with the regression analysis slope we adopt the same terminology for correlation, i.e. we refer to 'a correlation of $Y$ on $X$' simply to indicate the direction of the associated regressions analysis (although correlation values are between $X$ and $Y$). Errors in our observational estimates only influence the regression intercept values which we do not use. Regression slope uncertainty estimates are one standard error. Note, however, that these uncertainty estimates may not be reliable since they assume that the regression analysis residuals are normally distributed. For all linear regression analyses, it should be noted that the correlations and fractions of explained variance include any cross-correlations of both parameters with other variables. We present two-sided $p$ values ($p$) for the significance of the slopes of the linear fits with a null hypothesis that the slope is zero.

Simulated surface heat fluxes were analysed from 28 AMIP5 and 18 CMIP5 model experiments with suitable diagnostics and several recent Hadley Centre model simulations (see Supplementary Table 1). For AMIP5 and CMIP5, temporal averages over a common simulation period for all models of 1981–2005 inclusive were undertaken. Due to their different run periods, the Hadley Centre stand-alone atmospheric models (consistent with those employed in the coupled models) were averaged over the periods 1978–1995 for HadCM3-A[68] and HadGEM1-A[69], 1981–2005 for HadGEM2-A[70] (an AMIP5 model); and 1988–2008 for HadGEM3-GC2-A[71] and HadGEM3-GC3.1-A[48] (the Hadley Centre IPCC AMIP6 configuration). Observations and all models were interpolated onto a common HadGEM3-GC3-A N144 grid (a regular latitude-longitude grid with a resolution of approximately 80km at 55°).

Observational estimates were then subtracted from model fields to provide estimated model biases. Note that observational errors in our net, short-wave and long-wave downward heat flux observational estimates accumulate in our estimates of both simulated TAFB and total turbulent flux biases. There are also small ($\sim<2\,\mathrm{W\,m^{-2}}$) inconsistencies in this model-observational comparison of net and turbulent heat fluxes (due to our use of atmospheric surface energy flux observational estimates), including the heat flux associated with latent heat associated with snow; with latent heat associated with sea-ice volume changes over the averaging period; and with freshwater (precipitation minus evaporation) mass input[72]. The different averaging periods for models and observations also

introduce errors but these are expected to be small for Southern Ocean multi-annual means. For example, simulated differences for 40–60°S for 2001–2007 inclusive and 1988–2008 means were less than $0.5\,Wm^{-2}$ for all heat flux component terms and for net flux.

For the 34 CMIP5 models (including HadGEM2[70]), for which ocean temperature data were available, we estimated biases in SST, upper 300 m heat content and upper 1000 m heat content compared to objectively-analysed observations from EN4[32]. Heat content biases are expressed as layer mean temperatures. These simulations were taken from the historical runs and averaged over the period 1985–2005 and compared to EN4 averaged over the same period. For the additional Hadley Centre models SST biases were also estimated by subtracting the 1985–2004 EN4 mean. For HadCM3[68] and HadGEM1[69] bias estimates we used historical runs for periods and averaged over 1979–1996 and 1983–2008. For HadGEM3-GC3.1[48], the Hadley Centre CMIP6 configuration, and HadGEM3-GC2[71] biases were estimated from year 50–100 averages of present day control runs. We opted to use EN4 for SST to ensure consistency with the heat content analyses.

There are uncertainties in SST and ocean temperature observations, particularly associated with limited in-situ sampling and the lack of remotely sensed observations when cloud cover is high[32,73]. Observational SST errors do not affect our inferences from our correlations or regression slopes but do contribute to our area-mean and zonal mean SST bias estimates for individual models, presented in Fig. 1 and Fig. 6. For the period 1985–2005 the area-mean difference between EN4 and Reynolds[73] satellite-derived SST is 0.04 °C for 40–60°S (note that these two products are expected to suffer from common sampling issues). Zonal mean differences on our model grid between EN4 and Reynolds have maximum (mean) values between 35 and 65°S of 0.27 (0.09) K. These basic observational uncertainty estimates are considerably smaller than the mostly substantial SST biases in the models (see Fig. 1 and Fig. 6).

The CMIP5 and AMIP5 experiments provided an overlapping subset of 18 models for which we had both coupled and atmosphere-only ocean temperature and surface heat flux component diagnostics (Supplementary Table 1). Note that for historical simulations with Earth System Models observational estimates of greenhouse gas concentrations are prescribed. However, there may be some land-use differences compared to physical climate model simulations since land-use is simulated. Since there is no known justification for the exclusion of any of the models we chose to include all 18 models for which atmospheric heat flux components and ocean temperatures were available. However, note that the choice of models has an impact both on our correlation and regression analyses and the multi-model mean estimated biases (for details see the theoretical dependence of $\Delta T$ on $\Delta F_A$ and $\Delta C_O$ (R2) subsection).

For the 13 CMIP5 models with suitable diagnostics available (Supplementary Table 1), a linear regression was undertaken between SST and the Atlantic Meridional Overturning Circulation (AMOC) strength at its latitudinal maximum using a 70 year mean from the pre-industrial control simulations corresponding to years 1–70 of 1% $CO_2$ simulations. The pre-industrial control experiment was used to avoid contamination by different responses to forcing in the historical experiment. A 70-year mean was employed to minimise the impact of multi-decadal AMOC variability.

Westerly wind components at 10 m were analysed from CMIP5 historical simulations and AMIP5 simulations with suitable diagnostics (Supplementary Table 1). The southern hemisphere mid-latitude zonal wind maximum latitude (ZWML) was defined as the latitude of the maximum in the climatological annual zonal mean 10 m westerly wind component between 30°S and 70°S (southern hemisphere latitudes are defined as negative). Zonal mean westerly winds on the native model grids were interpolated onto a regular latitude grid with 0.075° spacing using a cubic spline interpolation to estimate the latitude of the zonal mean westerly wind maximum. Climatologies were defined over the period 1979–2005, which is the period of overlap with the modern satellite era, during which atmospheric reanalysis datasets are most reliable. We defined the bias in jet position relative to the ERA-Interim re-analysis[29] over the same 1979–2005 period (positive ZWML bias is northward). It should be noted that the ERA-Interim re-analysis winds are constrained by the assimilation of atmospheric wind, pressure, temperature and humidity observational estimates. A basic estimate of the observational uncertainty in ZWML of 0.15° is derived from the difference between ZWML estimates from the ERA-INTERIM and MERRA[66] re-analyses (Fig. 3).

**Ocean temperature composite analyses**. Zonal mean ocean temperature was calculated using the ocean potential temperature variable from the CMIP5 archive. Models with suitable diagnostics from the coupled historical climate simulation experiment (HIST) were evaluated. Zonal mean temperatures were calculated over the 'HIST' period 1979–2005 for each model. Of the models available, there were seven within the AMIP5 'high flux bias' (>5 $Wm^{-2}$) composite termed HIF ('CCSM4','bcc-csm1-1-m','CNRM-CM5','GISS-E2-R', 'ACCESS1.3', 'ACCESS1-0' and 'HadGEM2-CC') and seven in the AMIP5 'near zero or low flux bias' (<2.5 $Wm^{-2}$) composite termed NZF (IPSL-CM5A-MR, IPSL-CM5A-LR, CanESM2, MPI-ESM-LR, CMCC-CM, NorESM1-M and CSIRO-Mk3-6-0). Only the IPSL models have large negative net flux biases. Each model's zonal mean was interpolated onto a common regular one degree latitude, seventy depth level grid (mean vertical spacing ~100 m). The models in each ensemble were then averaged

together and the resulting HIF and NZF composites differenced to produce latitude-depth temperature difference fields. A paired t-test was conducted at each grid cell of the composites to assess if the mean of the elements of the two composites were significantly different from one another at the 95% level (making the assumption that the models are independent, which may not strictly be true). Those found to be significant were stippled in Fig. 4.

**Introduction to extended theory subsections**. The following seven subsections present additional details on our analyses of drivers of coupled model SST biases for readers with a particular interest in this topic. To facilitate the reader we duplicate material from the main manuscript.

First, we derive the equations governing a more complete analytical model, which includes the coupled response residual terms that do not depend linearly on local temperature. We then apply the more complete theory in four subsections corresponding to the R1 to R4 AMIP5/CMIP5 regression analysis relationships. The theoretical dependence of $\Delta T$ on $\Delta F_A$ and $\Delta C_O$ (R2) subsection includes more detailed discussion on the assumptions of independence of $\Delta F_A$ and $\Delta C_O$ and weak cross-correlations, and on the impacts of excluding outlying models. Next, we assess our equilibrium assumption and discuss the time-varying equations. Finally, we present three alternative simplified conceptual models to the one presented in the main manuscript and show that they are inconsistent with our AMIP5/CMIP5 ensemble results.

**Equilibrium mixed layer heat budget equations**. We can use the ocean mixed layer heat budget to derive an analytical solution for the equilibrium mixed layer temperature bias in any coupled model at any location as a function of the biases in the total heat fluxes into the mixed layer in stand-alone component atmospheric and ocean models (given suitable surface boundary conditions). Since SST is closely linked to mixed layer temperature, we assume they are approximately equal.

In the real world, at any location with a mixed layer of temperature, $T_{OBS}$, to conserve heat over any period, the time-derivative or tendency in the mixed layer heat content tendency, $dH_{OBS}/dt$, must be equal to the sum of the observed downward net surface heat flux, $F_{OBS}$, and observed combined horizontal and vertical ocean heat transport convergence in to the mixed layer, $C_{OBS}$, i.e.:

$$dH_{OBS}/dt = F_{OBS} + C_{OBS} \tag{M1}$$

Similarly, for any coupled model at any location over any period with a mixed layer temperature, $T$, the mixed layer heat content tendency, $dH/dt$, must be equal to the sum of the simulated surface heat flux, $F$, and simulated combined horizontal and vertical ocean heat transport convergence, $C$:

$$dH/dt = F + C \tag{M2}$$

Next we consider any coupled model location with a mixed layer temperature bias, $\Delta T = T - T_{OBS}$. Subtracting Eq. M1 from Eq. M2 and substituting for the simulated surface heat flux bias, $\Delta F = F - F_{OBS}$, and simulated ocean heat transport convergence bias, $\Delta C = C - C_{OBS}$, gives:

$$dH/dt - dH_{OBS}/dt = \Delta F + \Delta C \tag{M3a}$$

Approximate equilibration of the mixed layer with the atmosphere and upper-ocean typically occurs within a few decades for a coupled model spin up run. Once this has occurred, averaging over long multi-annual periods reduces the observed and simulated mixed layer heat content tendencies associated with climate variability. We show in a separate subsection below, on the equilibrium assumption, that for our multi-annual mean bias estimates from the CMIP5 historical simulations the heat content tendencies are small, i.e. the equilibrium assumption is valid. We therefore ignore the observed and simulated heat content tendencies, $dH/dt$ and $dH_{OBS}/dt$, in Eq. M3a to give:

$$\Delta F \approx -\Delta C \tag{M3b}$$

We can decompose $\Delta F$ into a stand-alone atmospheric model surface net flux bias, given realistic SST forcing, $\Delta F_A$ and a surface heat flux coupled response, including all feedbacks, $\Delta F_R$:

$$\Delta F = \Delta F_A + \Delta F_R \tag{M4}$$

We can approximate $\Delta F_A$ from the model's AMIP5 experiment net flux bias, if AMIP5 observed SST forcing is assumed to be realistic. The response term $\Delta F_R$ is associated with all local and non-local changes in oceanic and atmospheric states that occur only when the components are coupled. It is caused by the combined impact of all the model component errors. We can estimate $\Delta F_R$ as the difference between a model's CMIP5 net heat flux bias, i.e. atmospheric model surface heat flux error plus coupled feedbacks, and its AMIP5 net heat flux error, i.e.

atmospheric model surface heat flux error without coupled feedbacks. Hence, our decomposition in Eq. M4 becomes $\Delta F = \Delta F_A + (\Delta F - \Delta F_A)$.

Similarly, we decompose $\Delta C$ into a stand-alone ocean-ice model heat transport convergence error, $\Delta\Delta C_O$, given realistic atmospheric surface forcing, and a coupled feedback response term $\Delta C_R$, where $\Delta C_R = \Delta C - \Delta C_O$, since clearly $\Delta C = \Delta C_O + (\Delta C - \Delta C_O)$:

$$\Delta C = \Delta C_O + \Delta C_R \qquad (M5)$$

The coupled response $\Delta C_R$ (like $\Delta F_R$) is associated with local and non-local adjustments to oceanic and atmospheric states that result when model components are coupled from all of the model component errors, including AMIP5 surface momentum/freshwater forcing biases. Strictly, it should be noted that it is possible to have error cancellation within the $\Delta C_R$ term. For example, the coupled ocean response to atmosphere-only biases in momentum or freshwater fluxes could be compensated by other coupled adjustments. We are not able to estimate $\Delta C_O$ for any CMIP5 model so we are also unable to estimate $\Delta C_R$.

The Ocean Model Inter-comparison Project experiments (e.g. OMIP5)[24] uses stand-alone ocean-ice models, with prescribed best-estimate surface atmospheric forcing. In these models, obviously there can be no atmospheric adjustment to flux changes associated with SST biases. In principle, it might prove possible in the future to derive first order estimates of $\Delta C_O$ for a given CMIP5 model from its OMIP flux bias $\Delta F_O$ using the equilibrium assumption that $\Delta C_O \approx -\Delta F_O$. However, this requires further investigation and, unfortunately, there are currently insufficient (<10) OMIP experiments with consistent ocean components to those of the CMIP5 models to undertake regression analyses. Furthermore, for OMIP5, the assumption of realistic surface forcing might prove to be invalid, given large known errors in ocean forcing sets and questionable surface boundary condition assumptions.

Combining Eqs. M3b to M5 gives:

$$\Delta F_A + \Delta F_R \approx -(\Delta C_O + \Delta C_R) \qquad (M6)$$

$$\Delta F_A + \Delta C_O \approx -(\Delta F_R + \Delta C_R) \qquad (M7)$$

The combined stand-alone model component biases in surface heat flux and ocean heat transport convergence must therefore be almost exactly opposed by their associated combined coupled responses.

Let us suppose, as expected and already discussed, that $\Delta F_R$ depends linearly on the SST bias, $\Delta T$, with a negative sensitivity constant, $\lambda_F$, and a residual term, $R_F$. This residual term includes contributions from all remote errors and local errors, which do not depend linearly on SST biases. Although we cannot anticipate how $\Delta C_R$ relates to SST, let us suppose that $\Delta C_R$ also depends linearly on the mixed layer temperature bias, $\Delta T$, with a sensitivity constant, $\lambda_C$, and a residual non-linearly dependent term, $R_C$.

Hence, from Eq. M4:

$$\Delta F_R = \Delta F - \Delta F_A = \lambda_F * \Delta T + R_F \qquad (M8)$$

And from Eq. M5:

$$\Delta C_R = \Delta C - \Delta C_O = \lambda_C * \Delta T + R_C \qquad (M9)$$

Combining Eqs. M7 to M9 gives:

$$\Delta F_A + \Delta C_O \approx -(\lambda_F + \lambda_C) * \Delta T - (R_F + R_C) \qquad (M10)$$

For individual models, we can estimate $\Delta F_A$, $\Delta F$ and $\Delta T$ from our AMIP5/CMIP5 surface flux biases and CMIP5 SST biases, which allows us to estimate $\Delta F_R$. However, we cannot directly estimate $\Delta C_O$ or $\Delta C_R$. For individual models we also cannot estimate $\lambda_F$, $\lambda_C$, $R_F$ or $R_C$. We therefore cannot solve the equilibrium mixed layer budget for individual models. We can, however, combine Eqs. M1 to M10 to derive the expressions governing the R1 to R4 regression relationships defined in the main manuscript across the AMIP5/CMIP5 model ensembles.

To relate these expressions to the relationships between variations in our known parameters across the AMIP5/CMIP5 ensemble we must make several additional assumptions. We assume that $\Delta C_O$ and $\Delta F_A$ are uncorrelated so the unknown $\Delta C_O$ values simply contribute to the regression residual, limiting the correlation. We expect this assumption to be approximately valid since $\Delta C_O$ and $\Delta F_A$ are errors in completely different stand-alone ocean and atmospheric model components, which are largely developed separately. We also assume that the response sensitivity constants, $\lambda_F$ and $\lambda_C$, are approximately consistent across the models. We expect some deviations between individual model sensitivity constants, $\lambda_F$ and $\lambda_C$, from their multi-model mean values which will contribute to our residual $R_F$ and $R_C$ terms, together with other non-linear and non-local contributions to $\Delta C_R$ and $\Delta F_R$. If any of these regression residual terms were dominant we would not expect strong correlations for R1 to R4. Any regression result can be influenced by potentially

misleading cross-correlations of important external parameters with both of the variables used in the regression. We expect and assume these local and non-local cross-correlations to be weak. We discuss the potential implications of these assumptions on our inferences for drivers of coupled SST biases in the R2 subsection below.

In the following four subsections, we show that the expected R1 to R4 regression relationships from our more complete model and the associated inferences from the AMIP5/CMIP5 ensemble regression results are consistent with those from the general case model presented in the main manuscript.

**Theoretical dependence of $\Delta F_R$ on $\Delta T$ (R1).** Eq. M8 states:

$$\Delta F_R = \Delta F - \Delta F_A = \lambda_F * \Delta T + R_F \qquad (M8)$$

We expect $\lambda_F$ to be negative. Across a group of models, if there were consistency in their response sensitivity constants, $\lambda_F$, Eq. M8 would result in a negative correlation of surface heat flux coupled responses, $\Delta F_R$ on SST biases, $\Delta T$. The CMIP5 average response constant $\lambda_F$ would be given by the slope of this regression ($S_1$).

We find a strong negative correlation of $\Delta F_R$ on $\Delta T$ ($r_1 = -0.66$, $S_1 = -5.5 \pm 1.6 \, \mathrm{Wm^{-2}K^{-1}}$, $p = 2.8E\text{-}3$; R1). Our regression slope for $\Delta F_R$ on $\Delta T$, $S_1$, suggests that $\lambda_F \sim -5.5 \pm 1.6 \, \mathrm{Wm^{-2}K^{-1}}$. We use our estimate of $\lambda_F$ to estimate $R_F$ for use in the R3 subsection below. This will include a contribution from any variations in $\lambda_F$ across the models.

The CMIP5-AMIP5 flux component bias on SST bias, $\Delta T$, regression sensitivities are $-4.8 \pm 1.1$ and $-1.3 \pm 0.4 \, \mathrm{Wm^{-2}K^{-1}}$ for the total turbulent and long-wave flux, respectively. The sensitivity of $-1.3 \pm 0.4 \, \mathrm{Wm^{-2}K^{-1}}$ for long-wave suggests that a large fraction of the Stefan's Law estimated emitted long-wave $\sim 4.8 \, \mathrm{Wm^{-2}K^{-1}}$ is re-emitted towards the surface. The turbulent flux sensitivity of $-4.8 \pm 1.1$ is broadly consistent with observational estimates of its sensitivity for the Southern Ocean region derived from ERAI[35] of $< 10 \, \mathrm{Wm^{-2}K^{-1}}$.

**Theoretical dependence of $\Delta T$ on $\Delta F_A$ and $\Delta C_O$ (R2).** Re-arranging Eq. M10 to estimate $\Delta T$ gives:

$$\Delta T = -(\Delta F_A + \Delta C_O + R_F + R_C)/(\lambda_F + \lambda_C) \qquad (M11)$$

Equation M11 would be expected to give rise to correlations across the AMIP5/CMIP5 ensemble of SST biases, $\Delta T$, on either the stand-alone atmospheric model component surface heat flux biases, $\Delta F_A$, or stand-alone ocean model ocean heat transport convergence biases, $\Delta C_O$. We investigate the $\Delta T$ on $\Delta F_A$ regression relationship by assuming that $\Delta F_A$ and $\Delta C_O$ are uncorrelated. Note also that $R_F$ and $R_C$ are by definition uncorrelated with $\Delta T$. The ensemble multi-model mean ($\lambda_F + \lambda_C$) is given by minus the inverse of the $\Delta T$ on $\Delta F_A$ regression slope ($S_2$). The explained variance, $r_2^2$, is given by the variance in $\Delta F_A/(\lambda_F + \lambda_C)$ divided by the variance in $\Delta T$. Any variations in ($\lambda_F + \lambda_C$) will also contribute to the regression errors, limiting the size of the correlation.

We find a strong positive correlation for $\Delta T$ on $\Delta F_A$ (Fig. 1; $r = 0.84$, $S_2 = 0.10 \pm 0.02 \mathrm{KW^{-1}m^2}$, $p = 1.4E\text{-}5$; R2). This suggests that ($\lambda_F + \lambda_C$) $\sim -10.0$ ($-8.3$ to $-12.5$) $\mathrm{Wm^{-2}K^{-1}}$, assuming that $\Delta F_A$ and $R_F + R_C + \Delta C_O$ are independent as expected. Combining this estimate with our estimate from the R1 subsection of $\lambda_F \sim -5.5$ ($-3.9$ to $-7.1$) $\mathrm{Wm^{-2}K^{-1}}$ suggests that $\lambda_C \sim -4.5$ ($-1.2$ to $-8.6$) $\mathrm{Wm^{-2}K^{-1}}$. These large uncertainties result from combining the two regressions, but the central value of $\lambda_C$ suggests that it is of similar magnitude to $\lambda_F$. The correlation of 0.84 indicates that 70% of the variance in $\Delta T$ is explained by variations in $\Delta F_A$. If, as assumed, $\Delta F_A$ and $\Delta C_O$ are independent this implies that less than 30% of the $\Delta T$ variance can be explained by variations in $\Delta C_O$. We discuss the implications of each of our assumptions below.

We expect $\Delta F_A$ and $\Delta C_O$ variations across the models to be approximately independent since the stand-alone ocean and atmosphere models are separate models, which are largely developed independently. In fact, any tuning that might be undertaken once components are coupled to minimise local SST biases could, from Eq. M11, even be expected to introduce weak anti-correlation between $\Delta F_A$ and $\Delta C_O$. If significant, this anti-correlation would have reduced both our $\Delta T$ on $\Delta F_A$ correlation and the total of the sum of the variance in $\Delta T$ explained by $\Delta F_A$ and $\Delta C_O$ individually to less than one. Taking such an anti-correlation into account would make our estimate of 30% for the fraction of $\Delta T$ variance explained by $\Delta C_O$ even smaller. Anti-correlation would also reduce the magnitude of the regression slope, which would in turn increase the magnitude of our estimates of ($\lambda_F + \lambda_C$). Since $\lambda_F$ is estimated separately from R1 this would mean that our estimate of $\lambda_C$, which assumes no correlation, is an overestimate. Note, however, that this effect cannot be large, given that our $\Delta T$ on $\Delta F_A$ correlation coefficient is 0.84, i.e. fairly close to 1.0.

The choice of models influences the magnitude of the $\Delta T$ on $\Delta F_A$ correlation (and could also affect the $\lambda_F$ and $\lambda_C$ estimates). We chose to include all 18 models for which atmospheric heat flux components and ocean temperatures were available. There is no known justification for the exclusion of any of the models. Note, however, that the available models included two IPSL models, both of which had low net flux and SST biases compared to the other models. Their inclusion

increases the $\Delta T$ on $\Delta F_A$ correlation. For example, for AMIP5 net flux vs. CMIP5 SST linear regressions, the correlations ($p$ value) were 0.84 (1.4E-5), 0.77(2.7E-4) and 0.70 (2.6E-3) including both, one (IPSL-CM5A-MR) and no IPSL models, respectively. Hence, without the two IPSL models the fraction of SST bias variance explained by atmospheric model F biases reduces to around 50%. Spearman rank correlations are less sensitive to outliers than linear regressions. For AMIP5 net flux vs. CMIP5 SST regressions, the Spearman rank correlations (and $p$ values) were 0.83 (2.4E-05), 0.80 (1.3E-4) and 0.76 (7.1E-4) including both, one (IPSL-CM5A-MR) and no IPSL models, respectively.

Cross-correlations between both regression parameters and other important parameters can influence regression results, complicating inferences on causality. Our analysis considers the linear dependence of CMIP5 SST biases, a coupled ocean parameter, on AMIP5 net flux biases, a parameter from a stand-alone atmospheric model with no ocean model. Hence, any causality can only result from a dependence of CMIP5 SST on AMIP5 surface fluxes, not vice-versa. Clearly, we expect AMIP5 net flux biases to be correlated with all AMIP5 atmospheric model deficiencies that directly influence surface fluxes, e.g. cloud or boundary layer parameter biases. These biases would therefore influence SST biases through our proposed mechanism. Potentially misleading cross-correlation candidates are limited to any local or remote atmospheric model variables that influence AMIP5 surface heat, freshwater and momentum fluxes. Any such cross-correlation would only matter if these flux biases were to strongly influence area-mean CMIP5 SST biases through a different mechanism, e.g. such as by changing the coupled response term $\Delta C_R$.

Local 40–60°S area-mean correlations of AMIP5 net fluxes on AMIP5 precipitation minus evaporation freshwater and AMIP5 momentum fluxes are weak for our set of AMIP5/CMIP5 models ($r < 0.2$, $n = 17$). Furthermore, their weak correlations with SST ($r < 0.2$, $n = 17$) do not exceed those expected solely from their cross-correlation with net flux via our proposed mechanism. There is also no significant correlation between these fields and the regression residuals from the SST bias on net flux bias regression, i.e. there is no evidence of an independent physical link with SST. Note also that no significant correlation is evident for SST biases on sea surface salinity biases for 40–60°S. Clearly, we cannot rule out all potential non-local cross-correlations between heat, momentum and freshwater flux bias terms and 40–60°S heat flux biases. However, there is no physical basis to expect these remote terms to play an important physical role in the 40–60°S area-mean mixed layer heat budget.

**Theoretical dependence of ($\Delta F_R - R_F$) on $\Delta F_A$ (modified R3)**. Re-arranging Eq. M8 gives:

$$\Delta T = (\Delta F_R - R_F)/\lambda_F \tag{M12}$$

Substituting Eq. M12 in Eq. M11 for $\Delta\Delta T$, and re-arranging to evaluate $\Delta\Delta F_R - R_F$ gives:

$$\Delta F_R - R_F = -(\Delta F_A + \Delta C_O + R_F + R_C)\lambda_F/(\lambda_F + \lambda_C) \tag{M13}$$

This relationship combines those in subsections R1 and R2 and is therefore not independent of them.

Across a group of models, with consistent $\lambda_F$ and $\lambda_C$, Eq. M13 would be expected to give rise to correlations of $\Delta F_R - R_F$ on $\Delta F_A$, the atmospheric model surface heat flux biases. Again, it is necessary to assume, as expected, that $\Delta F_A$ and ($\Delta C_O + R_F + R_C$) are approximately uncorrelated. The slope of this regression ($S_3$) then provides an estimate for $-\lambda_F / (\lambda_F + \lambda_C)$. We find a strong negative correlation for modified R3 modified ($\Delta F_R - R_F$) on $\Delta F_A$ ($r = -0.84$, $S_3 = -0.54 \pm 0.09$, $p = 1.5$E-5). This provides an estimate of $\lambda_F/(\lambda_F + \lambda_C)$ of $0.54 \pm 0.09$, which is consistent with the values of $\lambda_F$ and $\lambda_C$ which we estimated from subsections R1 and R2, but appears to provide a slightly better constrained estimate of $\lambda_C/\lambda_F = 0.85$ (0.58 to 1.2). With our known parameters (i.e. $\Delta T$, $\Delta F_R$ and $\Delta F_A$), this relationship could potentially prove useful in other regions, e.g. where $\lambda_F$ is large but the standard deviation of $\Delta C_O$ is larger than that of $\Delta F_A$ so that the $\Delta T$ on $\Delta F_A$ correlation is weak.

**Theoretical dependence of $\Delta T$ on $\Delta F$ (R4)**. Two equivalent expressions may be derived for the coupled fluxes that are useful to interpret our $\Delta T$ on $\Delta F$ relationships:
From Eq. M8:

$$-\Delta C = \Delta F = \Delta F_R + \Delta F_A = \lambda_F * \Delta T + R_F + \Delta F_A \tag{M14}$$

Re-arranging gives:

$$\Delta T = (\Delta F - R_F - \Delta F_A)/\lambda_F \tag{M15}$$

From Eq. M9:

$$-\Delta F = \Delta C = \Delta C_R + \Delta C_O = \lambda_C * \Delta T + R_C + \Delta C_O \tag{M16}$$

Re-arranging gives:

$$\Delta T = -(\Delta T + R_C + \Delta C_O)/\lambda_C \tag{M17}$$

Useful expressions for the expected $\Delta T$ on $\Delta F$ regression relationship across an ensemble can be only derived when either $\Delta F_A$ or $\Delta C_O$ are small and can therefore be ignored (i.e. by ignoring $\Delta F_A$ in Eq. M15 or by ignoring $\Delta C_O$ in Eq. M17). The more general equation governing the $\Delta T$ on $\Delta F$ regression relationship depends on both of the stand-alone model component errors and the coupled responses. Substituting $z = \Delta C_O/\Delta F_A$ and combining Eqs. M10, M4 and M6, one can derive a general expression for the dependency of $\Delta T$ on $\Delta F$. However, this is not presented as it is hard to interpret since $z$ would be expected to vary randomly across the models. Furthermore, for 40–60°S, we find a weak positive correlation of $\Delta T$ on $\Delta F$ ($r = 0.35$, $p = 1.6$E-1) which does not permit useful quantitative analyses from its slope. These relationships also combine those in subsections R1 and R2 and are therefore not independent of them. However, with our known parameters ($\Delta T$, $\Delta F_R$ and $\Delta F_A$), Eq. M17 could potentially prove useful for other regions where $\Delta T$ on $\Delta F$ correlations are strong.

**The equilibrium assumption used in the theory**. The time-varying mixed layer budget equations for observations, a simulation and simulated biases are given by Eqs. M1, M2 and M3a, b, respectively. Since our study investigates simulated biases we consider Eq. M3a:

$$dH/dt - dH_{OBS}/dt = \Delta C + \Delta F$$

This time-varying budget equation can be used to derive a modified expression for $\Delta T$ (in a similar manner to that used to derive Eq. M11):

$$\Delta T = -(\Delta F_A + \Delta C_O + R_F + R_C - dH/dt + dH_{OBS}/dt)/(\lambda_F + \lambda_C) \tag{M18}$$

For any model, we can approximately estimate the tendency in the heat content of the mixed layer, $H$, over the time of the $\Delta T$ averaging period as follows:

$$dH/dt = \rho\, c_p d(DT)/dt \tag{M19}$$

where $D$ is the mixed layer depth, $c_p$ is the specific heat capacity of water, $\rho$ is the density of water and $T$ is the temperature of the mixed layer temperature and SST.

The observed heat content tendency ($dH_{OBS}/dt$) should not affect our regression slope or correlation estimates since its value is the same for all of the models. Similarly, any contribution to the simulated heat content tendency ($dH/dt$) from any multi-model mean simulated response to climate forcing common to all the models should have no impact. We expect the remainder of the simulated heat content tendency $dH/dt$ to vary approximately randomly across the models since it is should be primarily associated with climate variability. The validity of our equilibrium assumption therefore depends on the size of $dH/dt$ variations across the models relative to those of the known bias terms, i.e. $\Delta F_A$ and $\Delta F$ in Eqs. M18 and M3a. This will depend on the averaging period with longer averaging period tending to have smaller magnitude heat content tendencies.

For our AMIP5/CMIP5 simulations the $\Delta T$ averaging period is $\geq 25$ years. For the HadGEM2 historical run, the standard deviation of $dH/dt$ for 40–60°S, estimated from six different 20 year averaging periods, is 0.2 Wm$^{-2}$(for a 10 year averaging period it is 0.4 Wm$^{-2}$). This value of 0.2 Wm$^{-2}$ is clearly small compared to the AMIP5 net flux bias standard deviation of 6.6 Wm$^{-2}$ (~ 3%) and the CMIP5 net flux standard deviation of 3.7 Wm$^{-2}$ (~ 5%). It would also remain small even if mixed layer temperature tendencies associated with variability were several times larger in other models, which is not expected for an average over such a large region. Note also from Eq. M18 that this term would have acted as an additional noise term in our existing regression analyses, limiting the correlations. If this term were to be an important term in the budget, we would therefore not see such a strong CMIP5 SST bias on AMIP5 net flux bias correlation ($r = 0.84$).

Equations M18 and M19 can also be used to provide a first order estimate of the equilibration timescale for the mixed layer temperature, by making some rather crude assumptions. We assume that $D$, $\Delta F_A$, $\Delta C_O$, $R_F$, $R_C$, $\lambda_F$, $\lambda_C$, $\rho$, and $c_p$ are all approximately constant in time. We do not expect this assumption to be strictly valid, particularly for the $R_C$ and $\lambda_C$ coupled ocean response parameters that are expected to vary as the local and non-local ocean mixed layer temperature biases develop. Putting $X = \Delta T$ gives:

$$dX/dt = ((1/\rho c_p D) * (\Delta F_A + \Delta C_O + R_F + R_C) + X(\lambda_F + \lambda_C)) = a - bX$$

With $a = (\Delta F_A + \Delta C_O + R_F + R_C)/\rho c_p D$ and $b = -(\lambda_F + \lambda_C)/\rho c_p D$ so that $b$ is positive. Integrating in time and introducing a constant to ensure that $X$ is zero at time, $t = 0$ gives:

$$X = (-a/b)(e^{-bt} - 1) \tag{M20}$$

This provides a solution for $\Delta T$ which asymptotes to a stable equilibrium value with an e-folding timescale, $T_e$:

$$T_e = 1/b = \rho c_p D/(\lambda_F + \lambda_C)$$

Using approximate values of $c_p$ and $\rho$ of 4200 Jkg$^{-1}$ and 1000 kgm$^{-3}$, respectively, and dividing by 3600 x 24 to convert time units to days, gives a timescale $T_e$ of:

$$T_e \approx 49 \, D/|\lambda_F + \lambda_C| \text{ days}$$

We can apply this theory to investigate the time-evolution of the HadGEM3-GC2 SST biases. For approximate area-mean equilibration to occur most of the 40–60°S region must equilibrate, including any areas with particularly deep mixed layers. We therefore use the maximum annual mean mixed layer depth at any grid point within the 40–60°S region of 1000 m. As we do not know $|\lambda_F + \lambda_C|$ for HadGEM3-GC2 for the Southern Ocean, we assume the AMIP5/CMIP5 multi-model mean value of $|\lambda_F + \lambda_C|$ of 10 Wm$^{-2}$K$^{-1}$ approximately applies. This suggests the e-folding timescale should be around 13 years. There are considerable uncertainties in this estimate since we do know the regional mean $|\lambda_F + \lambda_C|$ value for HadGEM3-GC2 and the most appropriate mixed layer depth value to use is also uncertain (over most of the 40–60°S region the mixed layer depth is much shallower than 1000 m and the area-mean annual-mean mixed layer depth is only ~ 75 m).

In HadGEM2-GC2 present day control runs the Southern Ocean 40–60°S annual mean SST bias evolves over time with an e-folding timescale of around 8 years, asymptoting over around 20 years to a fairly stable mean value of around ~ 3 °C. The bias then persists for the remainder of the experiment (in this case ~ 200 years) with some smaller inter-annual fluctuations of less than approximately 1 K super-imposed. The simulated HadGEM3-GC2 e-folding time scale of ~ 8 years is therefore broadly consistent with our rather uncertain and crude theoretical estimate of equilibration timescales of ~ 13 years.

The Southern Ocean SST bias is evident at a wide range of forecast timescales across all of the systems in the Met Office traceable seamless forecasting suite which employ the HadGEM3-GC2 model. This includes forecasts of a few days from the coupled Numerical Weather Prediction system; forecasts of months to years from Seasonal and Decadal forecasting systems; and forecasts of many centuries in centennial climate prediction experiments[74]. This basic theory provides a potentially useful framework to try to link certain biases in coupled numerical weather prediction and seasonal/decadal forecasting systems to equilibrated biases in long term climate simulations for models. Of course, this would only be useful for model regions where the bias evolves as a negative exponential which asymptotes towards a long term value such as the Southern Ocean[74]. Note, however, that for many model regions the time evolution of the bias in seasonal/decadal forecast is considerably more complicated[74] suggesting our crude theoretical assumptions may well not apply in these model regions.

**Additional theoretical simplified conceptual case models**. This subsection considers three additional simplified conceptual solutions to Eqs. 1b and 2 in which either or both of the ocean terms are assumed to be dominant. We contrast the expectations from these models with those from the simplified conceptual case model introduced in the main manuscript in which the ocean terms are assumed to be small. We demonstrate that our AMIP5/CMIP5 ensemble Southern Ocean 40–60°S regression results are inconsistent with the expectations from these new cases but are broadly consistent with the case when ocean terms are assumed small. Note however, that it is possible that these alternative simplified conceptual cases might be consistent with AMIP5/CMIP5 ensemble results for other regions.

Eq. 1b is (see main manuscript for definitions of terms):

$$\Delta F_A + \Delta C_O \approx -(\Delta F_R + \Delta C_R) \tag{1b}$$

To recap, the AMIP5/CMIP5 ensemble regression relationship results are: (R1) a strong negative correlation ($r = -0.66$) of CMIP5-AMIP5 net flux biases on SST biases, i.e. $\Delta F_R$ on $\Delta T$; (R2) a strong positive correlation ($r = 0.84$) of CMIP5 SST biases on AMIP5 net flux biases, i.e. $\Delta T$ on $\Delta F_A$; (R3) a strong negative correlation ($r = -0.84$) of CMIP5-AMIP5 net flux biases on AMIP5 net flux biases, i.e. $\Delta F_R$ on $\Delta F_A$; and (R4) a weak correlation ($r = 0.35$) of CMIP5 SST biases on CMIP5 net flux biases, i.e. $\Delta T$ on $\Delta F$ (where $\Delta F = \Delta F_A + \Delta F_R$).

The four conceptual cases have different expected signatures across the model ensemble for each of these four regression relationships. All four cases ignore remote and indirect contributions to the coupled response terms that do not depend on local SST (i.e. we assume $\Delta F_R \approx \lambda_F \Delta T$ and $\Delta C_R \approx \lambda_C \Delta T$). If important,

these contributions will act as additional error terms reducing the correlations in our expected regression results. The expected results for the R1 to R4 regressions for each of the cases are detailed below, together with their agreement with the AMIP5/CMIP5 ensemble results.

In case 1 (see main manuscript and Fig. 2c), atmospheric model surface heat flux biases drive local SST biases through balancing surface heat flux response, i.e. small $\Delta C_O$ and $\Delta C_R$ so $\Delta F_R \approx \lambda_F \Delta T \approx -\Delta F_A$. The atmospheric model provides surplus heat into the surface which causes the local SST to warm resulting in a compensating positive surface heat loss response (or vice-versa). We expect for R1 a strong negative correlation of $\Delta F_R$ on SST biases; for R2 an approximately compensating strong positive correlation of SST biases on $\Delta F_A$; for R3 a strong negative correlation of $\Delta F_R$ on $\Delta F_A$; and for R4 no correlation of SST biases on coupled net flux difference ($\Delta F$), since $\Delta F = \Delta F_R + \Delta F_A \approx 0$. Our results are broadly consistent with all of these expectations.

In case 2, atmospheric model surface heat flux biases drive local SST biases through ocean heat transport convergence balancing response, i.e. small $\Delta C_O$ and $\Delta F_R$ so $\Delta C_R \approx \lambda_C \Delta T \approx -\Delta F_A$. The atmospheric model provides surplus heat into the surface causing the local SST to warm resulting in a compensating negative ocean heat transport convergence response (or vice-versa). We expect for R1 no correlation of $\Delta F_R$ on SST biases (and a negative correlation of $\Delta C_R$ on SST biases which we can't test); for R2 a strong positive correlation of SST biases on $\Delta F_A$; for R3 no correlation of $\Delta F_R$ on $\Delta F_A$; and for R4 strong positive correlation of SST biases on coupled net flux bias ($\Delta\Delta F$). Our results contrast with expectations for R1, R2 and R3, ruling out this case.

In case 3, ocean model heat transport convergence biases drive local SST biases through balancing surface heat flux response, i.e. small $\Delta F_A$ and $\Delta C_R$ so $\Delta F_R \approx \lambda_F \Delta T \approx -\Delta C_O$. The ocean model produces surplus ocean heat transport convergence causing the local SST to warm resulting in a compensating positive surface heat loss response (or vice-versa). We expect for R1 strong negative correlation of $\Delta F_R$ on SST biases; for R2 no correlation of SST biases on $\Delta F_A$; for R3 no correlation of $\Delta F_R$ on $\Delta F_A$; and for R4 strong negative correlation of SST biases on coupled net flux biases, $\Delta\Delta F$. Our results contrast with expectations for R2, R3 and R4, ruling out this case.

In case 4, ocean model heat transport convergence biases drive local SST biases through balancing ocean heat transport convergence response, i.e. small $\Delta F_A$ and $\Delta F_R$ so $\Delta C_R \approx \lambda_C \Delta T \approx -\Delta C_O$. The ocean model produces surplus ocean heat transport convergence, causing the local SST to warm, resulting in a compensating negative ocean heat transport convergence response (or vice-versa). We expect for R1 no correlation of $\Delta F_R$ on SST biases (and a negative correlation of $\Delta C_R$ on SST biases which we can't test); for R2 no correlation of SST biases on $\Delta F_A$; for R3 no correlation of $\Delta F_R$ on $\Delta F_A$; and for R4 no correlation of SST biases on coupled net flux difference, $\Delta F$. Our results contrast with expectation R1, R2 and R3, ruling out this case. Note that an additional case when all of the local terms in Eq. 1b are small, i.e. both the local component errors are small, and neither response depends on local SST so local SST biases primarily result from non-local or momentum/freshwater forcing component biases could also give this result.

In summary, to support our analyses in the main manuscript we have considered 4 simple conceptual cases in which pairs of component error and response terms in Eq. 1 are assumed small. Our four regression results for the AMIP5/CMIP5 ensemble for 40–60°S rule out cases 2 to 4 but are broadly consistent with case 1 (the conceptual case model presented in the main manuscript), where the ocean model C bias ($\Delta C_O$) and C response terms ($\Delta C_R$ and $\lambda_C$) are assumed small.

**Code availability**. Due to intellectual property right restrictions, we cannot provide the source code or the documentation papers for HadGEM3-GC2 or GC3.1. The Met Office Unified Model (MetUM) is available for use under licence. A number of research organisations and national meteorological services use the MetUM in collaboration with the Met Office to undertake basic atmospheric process research, produce forecasts, develop the MetUM code and build and evaluate Earth system models. For further information on how to apply for a licence, see http://www.metoffice.gov.uk/research/collaboration/um-partnership.

## Data availability

The AMIP5/CMIP5 data are available from the Coupled Model Inter-comparison Project archive. The HadGEM3-GC2 and HadGEM3-GC3.1 present day control experiment and associated stand-alone atmosphere experiment data are available from Met Office on reasonable request. All of the observational datasets are publically available (see associated references for details).

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

## Acknowledgements

Met Office authors were supported by the Joint UK BEIS/Defra Met Office Hadley Centre Climate Programme (GA01101) and Public Weather Service. The UK National Environment Research Council (NERC) Projects/Programmes DEEP-C NE/K005480/1 (R.A., C.L., P.H., J.G.), HOSTACE NE/J020788/1 (D.B.), SMURPHS NE/N006054/1 (R.A., C.L., S.J., J. G.), ORCHESTRA NE/N018095/1 (A.M., P.H., S.J.) and the BAS core science programme Polar Science for Planet Earth (T.B.) funded or supported author contributions. MM was funded by Austrian Science Fund project P28818. Malcolm Brooks undertook HadGEM3-GC3-A surface albedo scheme developments. Dave Storkey, Daley Calvert and Tim Graham contributed to HadGEM3-GC3 ocean model developments. We thank Sean Milton, Matt Palmer, Colin Jones, Till Kuhlbrodt, Keith Haines, Mike Bell, Phillip Brohan and Maria Valdivieso for useful discussion. We acknowledge the World Climate Research Programme's Working Group on Coupled Modelling, which is responsible for CMIP, and we thank the climate modelling groups (listed in Supplementary Table 1 of this paper) for producing and making available their model output. For CMIP the U.S. Department of Energy's Program for Climate Model Diagnosis and Inter-comparison provides coordinating support and led development of software infrastructure in partnership with the Global Organization for Earth System Science Portals. We are grateful to Sarah Gille, Steven Seims and two anonymous reviewers for useful comments over several reviews that led to many improvements to this manuscript (since its original submission to Nature Climate Change in November 2016).

## Author contributions

P.H., T.B., A.M., J.G., A.B. and C.R. undertook the AMIP5/CMIP5 model analyses. S.J., R.A., C.L., J.E., M.M. and D.B. provided observational heat flux estimates and/or associated guidance. P.H., K.W., J.E., D.C., J.M., P.F., K.F., C.S., S.H., J.R., L.T., H.H., R.W., S.B. contributed to the process evaluation and developments to HadGEM3. P.H., R.A., J.G., R.W., J.E., H.H., J.M., K.W., A.B., P.F., A.M., T.B., S.J., M.M., D.B. contributed to the manuscript.

## Additional information

**Competing interests:** The authors declare no competing interests.

