## [Peer Review File · Nature Communications]

Reviewer #1 (Remarks to the Author):

I appreciated the opportunity to re-review this manuscript. The revised manuscript has been expanded and more clearly explains the methodology. The manuscript presents interesting ideas that will be valuable to the community. I'm still troubled by a number of details, and I think further clarification would be in order.

(1) The initial assumption of the paper is that over the 30-year period considered in this study, air-sea fluxes plus ocean convergence $(F+C) = H \, dT/dt = 0$ (where H is mixed-layer depth). The notion that $dT/dt=0$ is testable in models. In the ocean itself, it's not justified---numerous studies have pointed out long-term temperature trends in SST. The "convergence" should be diagnosable, at least in the CMIP models, and that would help justify the basic balance underlying this study.

(2) The data products used in this analysis are assumed to be reliable enough to derive further quantities from them. The analysis is carried out for the time period from 1985 to 2005, which essentially pre-dates Argo in the Southern Ocean. For most of the 1985-2005 time period, infrared SST is the leading source of temperature information. Infrared SST is available only in cloud-free conditions, so there are potential clear-sky and seasonal biases in the SST product and in everything derived from SST. There's also little certainty that the observed radiative fluxes achieve sufficient accuracy to derive further information from them. In particular, given the clear-sky bias of the SST, it's hard to have confidence in SST-based inferences about the role of clouds.

(3) The manuscript assumes that since 70% of the delta T variance is explained by Delta F, then only 30% can be explained by convergence errors within the ocean model. This relies on an assumption that flux errors and convergence errors are orthogonal. In reality, errors could be correlated, and it seems to me that a "partial correlation" framework would be helpful. The fact that fluxes explain 70% of the variance, doesn't tell us that oceanic processes won't also explain 70% of the variance.

(4) The conclusions of the manuscript are built on the assumption that correlation demonstrates causation. The authors have identified correlations between a number of quantities, but without a mechanistic analysis of model dynamics, the authors should not be asserting that the results

represent a causal relationship and not a correlation linked to some other mechanism that happens to influence both of the correlated fields.

(5) For an assessment of this sort, I think a full assessment of the oceanic convergence would be in order. That would involve analyzing the ocean component of the CMIP models, and perhaps an ocean state estimate, such as SODA or MERCATOR. But it should not be an insurmountable hurdle.

(6) More detailed assessment of the models would be warranted. If one were to carry out this analysis using a CMIP5 model as truth, rather than observations, would the same conclusions emerge?

(7) The presentation is still quite difficult. Although the manuscript is built on the notion that $dT/dt = 0$, it never actually uses dT/dt . The manuscript talks extensively about convergence, but the usage is specific to vertical exchange of heat across the base of the mixed layer (rather than horizontal convergence.) The equations underpinning this analysis could be presented more clearly.

Reviewer #2 (Remarks to the Author):

This is an ambitious effort. The authors fully recognise that there exist “large uncertainties in conventional observational estimates of surface heat fluxes”, but then proceed to construct net downward surface fluxes from TOA satellite observations (CERES EBAF) and ERA-Interim reanalysis.

I believe a more comprehensive statement on the uncertainty/limitations the surface fluxes is necessary, and that it needs to be repeated in the conclusions.

Using a different reanalysis product and different surface radiation flux estimates will potentially alter the regression relationships and ensuing conclusions dramatically. Is it possible to estimate the sensitivity of these correlations to the ‘observational’ surface fluxes? I am uncomfortable simply stating that “accurate quantitative estimates of flux component uncertainties are not currently achievable” and then proceeding with the analysis.

I also contest that “these atmospheric model net flux errors are primarily due to biases in short-wave radiation.” My latest understanding is that large biases may exist in the zonal distribution of precipitation over the SO. You can argue that precipitation is part of the LH + SH flux, but when stated as a SW bias, one immediately thinks of supercooled liquid water and radiative transfer, not

precipitation. Calling it simply a SW radiation bias is then wrong. I respect, however, that the SW radiation bias is in the literature.

If, however, we take this initial leap of faith in the surface fluxes, then model development and conclusions are very interesting.

I will state that I 'still' found the primary text difficult to fully appreciate/follow. From the earlier reviews, I note this has been previously discussed and addressed. It was essential for me to read the supplemental material. I respect space is limited, but certainly moving forward some of this material should be considered.

Steve Siems

2) Point by point responses to individual reviewer comments

Reviewer #1 (Remarks to the Author):

I appreciated the opportunity to re-review this manuscript. The revised manuscript has been expanded and more clearly explains the methodology. The manuscript presents interesting ideas that will be valuable to the community. I'm still troubled by a number of details, and I think further clarification would be in order.

- Thank you again for your considerable effort and persistence in reviewing our manuscript, which we very much appreciate. Your many comments have considerably helped us to develop and clarify our paper, particularly to ensure the explanation of the new theory is defensible and comprehensive. Please note again, however, that the theory is not the main focus of the paper. After a positive initial review in Dec 2016, it was added in response to reviewer requests to provide a plausible physical explanation for our interesting strong CMIP5 SST bias on AMIP5 net flux bias correlation result. The theory text obviously needs to be clear and robust so we very much appreciate your efforts to help with this. We hope this has now been achieved but if you feel further changes remain necessary, we would welcome specific comments on text to allow us to clarify, remove or caveat individual paragraphs, sentences or inferences. In addition to our responses to each of your points below, we have also marked up the word document manuscript with changes associated with each point.

R1.1) The initial assumption of the paper is that over the 30-year period considered in this study, air-sea fluxes plus ocean convergence $(F+C) = H dT/dt = 0$ (where H is mixed-layer depth). The notion that $dT/dt=0$ is testable in models. In the ocean itself, it's not justified---numerous studies have pointed out long-term temperature trends in SST. The "convergence" should be diagnosable, at least in the CMIP models, and that would help justify the basic balance underlying this study.

- Thank you for raising this point, which needed a new statement. The observational mixed-layer or full column equilibrium assumption (heat content tendency dH_{OBS}/dt is small compared to flux) is often used in the literature, e.g. when deriving implied ocean heat transport for multi-annual means. Since the magnitude of the Southern Ocean regional mean net flux is thought to be fairly small it is natural to question the validity of this assumption. Since we consider simulated biases, however, the validity of the mixed layer equilibrium assumption depends on the relative size of the variations across the models in the mixed layer heat content tendency dH/dt compared to the fairly substantial variations across the models in coupled and atmosphere only net flux biases. In a dedicated new sub-section in the Methods on the equilibrium assumption, which considers the time varying equations, we have demonstrated that this is a very small term compared to the other terms in the budget. We also explain that, if significant, this term would have primarily acted as an additional error term in our regressions that would have weakened our correlations. If any of our assumptions were not valid we would not see such a strong correlation ($r=0.84$). In the Methods, we also present the time-varying mixed layer heat budget equations, as requested in your comment below (R1.7). We also derive a simple analytical solution to these equations with some crude assumptions to provide a first order estimate of the equilibration timescale.

This comment is discussed in detail in point (b) of our initial response letter.

R1.2) The data products used in this analysis are assumed to be reliable enough to derive further quantities from them. The analysis is carried out for the time period from 1985 to 2005, which essentially pre-dates Argo in the Southern Ocean. For most of the 1985-2005 time period, infrared SST is the leading source of temperature information. Infrared SST is available only in cloud-free conditions, so there are potential clear-sky and seasonal biases in the SST product and in everything derived from SST. There's also little certainty that the observed radiative fluxes achieve sufficient accuracy to derive further information from them. In particular, given the clear-sky bias of the SST, it's hard to have confidence in SST-based inferences about the role of clouds.

- The observational reference makes no difference to our correlation or regression slope estimates. This is discussed in detail in point (a) of the initial response letter below (we have duplicated the explanatory figure below).

Fig 1 – The CMIP5 SST bias on AMIP5 net flux bias regression lines and points for hypothetic perfectly correlated SST and net flux bias distributions with a zero intercept (blue line). Also included are the regression line for: (a) a hypothetic correction of 5 Wm^{-2} to the net flux observational estimate (green line); and (b) a hypothetic correction of 0.5K to the SST observational estimate (red line). The correlation and regression slopes are identical for all three series (only the regression intercept changes).

Our inferences on the link between CMIP5 SST biases and AMIP5 cloud errors are based on our regression results so are therefore independent of the observational accuracy. As stated, the observational uncertainties could potentially influence our inferences on flux and SST biases in individual models. For SST, this only applies to our discussion of the improvements to HadGEM3 SST biases. We state in the Methods that EN4 and Reynolds SST agree within 0.04°C . This difference is small compared to substantial $>1\text{-}3^\circ\text{C}$ model zonal or area annual mean SST biases but we accept and now state that both products may suffer from the same sampling issues you mention.

We have now added a new subsection in the main manuscript on the observations including a concise statement on our estimates of observational uncertainty and referring to the methods. We quote these uncertainties in the captions of Figs 1 and 3 but explain in another new subsection on regression analyses why they do not affect our regression results. We also include more detailed statements in our Methods where we also note the cloud-related sampling issues you describe. We do not expect these sampling errors to introduce potentially larger seasonal biases in these multi-annual mean observational estimates since for most ocean temperature products to avoid seasonal aliasing anomalies are derived monthly (relaxing towards climatology for days with missing data) before being averaged to estimate the annual means.

Note also that our conclusions on model improvements associated with cloud-related parameterisations are primarily based on the Hadley Centre atmospheric model heat flux component bias analyses in which SST is prescribed. The improved estimated SST biases in HadGEM3-GC3.1 relative to HadGEM3-GC2 resulting from improved atmospheric model net flux biases represent additional independent evidence for the link between clouds, radiation and SST biases (see our response to R1.4).

We have also extended our consideration of uncertainties more generally, including a new table (Table 2) on the changes in the Hadley Centre model hierarchy and added estimates of zonal mean observational uncertainties to accompany Fig 6. We have also removed the TAFB sub-panel from this figure.

R1.3) The manuscript assumes that since 70% of the delta T variance is explained by Delta F, then only 30% can be explained by convergence errors within the ocean model. This relies on an assumption that flux errors and convergence errors are orthogonal. In reality, errors could be correlated, and it seems to me that a "partial correlation" framework would be helpful. The fact that fluxes explain 70% of the variance, doesn't tell us that oceanic processes won't also explain 70% of the variance.

- We agree that strictly the total explained variance could potentially be up to 2 for the sum of two fully correlated variables or less than 1 if they are anti-correlated. However, we state that our inference assumes that ΔF_A and ΔC_O are independent. Without knowing ΔC_O the effect of this cross-correlation is impossible to quantify since it depends on both the relative sizes of variations across the model in ΔF_A and ΔC_O and any correlation between them.

As we now state in the main text, we expect stand-alone atmosphere and stand-alone ocean model flux errors to be approximately independent since they are errors associated with completely separate model components, which are largely developed and evaluated independently. Tuning these model components in a coupled model to minimise coupled SST might potentially be expected to introduce weak anti-correlation, i.e. error cancellation between the atmosphere and ocean components. Any anti-correlation would have reduced both our CMIP5 correlation ΔT on AMIP5 ΔF_A correlation and the total explained variance (r^2) by ΔF_A and ΔC_O to less than one. This would result in an even smaller estimate of the ocean model error contribution. We now discuss this in more detail in the extended theory subsection on the R2 ΔT on ΔF_A regression.

R1.4) The conclusions of the manuscript are built on the assumption that correlation demonstrates causation. The authors have identified correlations between a number of

quantities, but without a mechanistic analysis of model dynamics, the authors should not be asserting that the results represent a causal relationship and not a correlation linked to some other mechanism that happens to influence both of the correlated fields.

- We obviously agree that correlation alone does not demonstrate causation. However, in combination with theory it can provide very strong evidence. For example if one knows $A=B+C$ and there is a strong correlation of C against A, it suggests that variations in A are the dominant driver of variations in C. Our theory shows that both atmospheric and ocean model stand-alone heat flux errors are important. However, the regression shows us that the variations in atmospheric model F errors appear to be the dominant driver of SST variations across the AMIP5/CMIP5 models. See also point (d) our initial response letter below and the statement from Reviewer 3 at the first review. We have carefully caveated our conclusions to ensure they reflect the evidence discussed below.

We agree that cross-correlations can influence correlations. In fact, we make this exact point about previous inferences from correlations between CMIP5 SW and CMIP5 SST. The use of AMIP5 and CMIP5 greatly improves the ability to demonstrate causality, as the original Reviewer 3 acknowledged (see quote in our initial response letter). This is because the use of AMIP5 for one of our regression variables also considerably reduces the potential for cross-correlation since coupled feedbacks are not included in AMIP5 since SST is prescribed. CMIP5 parameters cannot cause AMIP5 parameters since AMIP5 is a reduced-complexity version of CMIP5.

Clearly, through our proposed mechanism there will be a dependence of AMIP5 surface net and component fluxes on other AMIP5 atmospheric driving variables. As we state, the primary control on net surface downward short-wave has been shown to be cloud characteristics, e.g. particularly cloud amount and brightness that are deficient in the models. The net downward long-wave, sensible and latent heat flux components have multiple controls, including clouds and boundary layer characteristics. We have now included this point at the start of the sub-section on individual heat flux components in the theory section of the main text.

To influence the CMIP5 ocean any other more spurious and potentially misleading AMIP5 cross-correlated atmospheric driving variable would have to change surface heat, momentum or freshwater fluxes. These surface flux terms would in turn have to play an important role in the mixed layer heat budget. Within the theory, there is scope for local or remote momentum and freshwater fluxes and non-local heat fluxes to influence the mixed layer heat budget through the ocean C coupled response term. However, we can check the magnitude of these potential cross-correlations at least locally by investigating correlations of area-mean net flux and SST biases on area-mean momentum and freshwater flux biases. We find these local cross-correlations to be very weak. We now state this in the main text and discuss it in more detail in the extended theory subsection on the R2 CMIP5 SST bias on AMIP net flux bias correlation in the Methods.

Note also that our conclusions on HadGEM3 model improvements associated with cloud-related parameterisations are primarily based on the AMIP heat flux component bias analyses. The improvement in SST biases in HadGEM3-GC3.1 that occurred when HadGEM3.1-A atmospheric model net flux biases were improved therefore represents additional independent evidence for the link between clouds, radiation and SST biases. We now state this in the main text.

R1.5) For an assessment of this sort, I think a full assessment of the oceanic convergence would be in order. That would involve analyzing the ocean component of the CMIP models, and perhaps an ocean state estimate, such as SODA or MERCATOR. But it should not be an insurmountable hurdle.

- Since the mixed layer heat content tendency term variations are small compared to net flux bias variations we could in principle estimate ΔC from $-\Delta F$ to the necessary accuracy. However since, as we state, we do not have sufficient consistent ocean-only models (none have adequate surface forcing) to estimate the ΔC_O terms, we could not anyway make use of estimates of ΔC , i.e. estimating ΔC would not help since we anyway could not estimate ΔC_R which is needed to better constrain our estimate of λ_C . In our opinion, any direct estimates of C from observations or simulations would be unlikely to be reliable since heat transport convergences are extremely sensitive to noise/errors (although we should again emphasize that observational errors do not affect our regression inferences). We have clarified these statements in both the Main Text and in the extended theory in the Methods. This is also discussed in detail in point (c) of our initial response letter below.

R1.6) More detailed assessment of the models would be warranted. If one were to carry out this analysis using a CMIP5 model as truth, rather than observations, would the same conclusions emerge?

- Our regression results would be identical with a different observational or simulation reference or even no reference, i.e. using the underlying parameter rather than its bias estimates. This is because the parameter mean is removed in estimation of correlation or regression slope. Their values therefore only relate to the bias variations across the models, which we know. The multi-model means are affected by the references but these only affect the regression intercepts, which we do not use. This was concisely stated in the introduction. However, since this point has caused confusion for two reviewers it obviously needed to be explained in more detail to help to clarify it. We have now re-organised, expanded and clarified the introductory text including new subsections on our observational products and their uncertainties and the regression analyses. In the latter section we now very clearly state that there is no sensitivity of our regression results to the accuracy of our observational products or any other reference such as a model. Wherever the accuracy of the observations matter, i.e. the consideration of individual model biases, and related Figs, we state their uncertainties and caveat our inferences accordingly. We have also separated our conclusions into those that depend on the accuracy of the observational estimates and those that do not.

R1.7) The presentation is still quite difficult. Although the manuscript is built on the notion that $dT/dt = 0$, it never actually uses dT/dt . The manuscript talks extensively about convergence, but the usage is specific to vertical exchange of heat across the base of the mixed layer (rather than horizontal convergence.) The equations underpinning this analysis could be presented more clearly.

- Thank you again for these useful comments. We have now moved the extended theory, which presents a detailed derivation of our equations, from Supplementary Material to the Methods. We hope that the inclusion of this extended text in the manuscript Methods will reduce the need to document non-essential detail in the main text theory section so we can maintain the overall focus on our interesting results. We have also included the time-varying equations in the Methods extended theory sub-sections on governing equations and a new

sub-section on the equilibrium assumption. We have also clarified and where necessary extended the main text theory sub-section throughout and where necessary refer to the extended theory in the Methods.

The ocean heat transport convergence, C , into the mixed layer is the combined vertical and lateral convergence terms. Unfortunately, the full description was removed at the last submission to try to simplify the text (it was described in the then Supplementary Material and indicated in Fig 1a). However, we have now again included this full description. Thank you for pointing this out.

Reviewer #2 (Remarks to the Author):

- Thank you Dr Siems for these useful comments, which we very much appreciate. In addition to our responses to each of your points below, we have also marked up the word document manuscript with changes associated with each point.

R2.1) This is an ambitious effort. The authors fully recognise that there exist “large uncertainties in conventional observational estimates of surface heat fluxes”, but then proceed to construct net downward surface fluxes from TOA satellite observations (CERES EBAF) and ERA-Interim reanalysis. I believe a more comprehensive statement on the uncertainty/limitations the surface fluxes is necessary, and that it needs to be repeated in the conclusions.

- As stated in the main text and discussed in detail in the Methods, we agree that there are large uncertainties in heat flux products that cannot be very accurately quantified. In the Methods, we also explained why our new observational method to estimate fluxes helps to constrain net flux estimates (e.g. see Supplementary Fig 5). We have now added a sentence making this point to a new section of the Main Text on our observational products and their uncertainties. In this section, we now also explain why we opt to estimate the turbulent flux as a residual referring to two papers on air-sea flux uncertainties (we removed these in earlier submission due to reference number limitations in Nature Climate Change):

Bourassa, M. A. *et al.* High-latitude ocean and sea-ice surface fluxes: challenges for climate research. *Bull. Am. Meteorol. Soc.*, **94**, 403-423 (2013).

Smith, S. R., Hughes, P.J. and Bourassa M. A. A comparison of nine monthly air-sea flux products. *Int. J. Climatol.*, **31**, 1002-1027 (2011).

The observational uncertainties make no difference to our inferences from the regressions as we explain in point (a) of our initial response letter and the responses to R1.2 and R1.6 above. We have re-arranged and clarified the introductory material in two new subsections on our observational products and the regression analyses. We now discuss observational uncertainties before explaining in detail why they do not matter for the regression inferences. We have also now separated our conclusions into those that depend on the accuracy of our observational estimate and those that do not.

To improve their accuracy we have also now modelled the TAFB uncertainties rather than using the analytical estimate. We include text explaining how this was undertaken in the Methods although it does not change the value of the TAFB uncertainty estimate. We have added Table 2 that presents the changes in Hadley Centre Models and CMIP5 multi-model

means and standard deviations together with the observational uncertainties. We also state which heat flux and SST changes between HadGEM3-GC2 and HadGEM3-GC3.1 appear to be improvements. We have also added estimates of zonal mean flux component and SST uncertainties in the Methods and quote the numbers on Fig 6 (we have removed the TAFB sub-panel). Wherever the observational uncertainties are relevant they are now quoted, i.e. the interpretation of flux biases for individual models (e.g. Table 2, Figs 5 and 6). We also interpret our results in the context of our uncertainty estimates (e.g. Table 2), which we acknowledge are fairly basic by necessity.

R2.2) Using a different reanalysis product and different surface radiation flux estimates will potentially alter the regression relationships and ensuing conclusions dramatically. Is it possible to estimate the sensitivity of these correlations to the ‘observational’ surface fluxes? I am uncomfortable simply stating that “accurate quantitative estimates of flux component uncertainties are not currently achievable” and then proceeding with the analysis.

- This is clearly a confusing issue. Our regression slope and correlation results and inferences do not depend on the observational reference or its accuracy as explained in point (a) of our initial response letter and the responses to R1.2 and R1.6 above. We have extended and rearranged the introductory text with two new subsections on our observational products and their uncertainties and our regression analyses. Where uncertainties are important we have interpreted our results in the context of the uncertainties and added caveats to our conclusions (as detailed in our response to point R2.1 above).

R2.3) I also contest that “these atmospheric model net flux errors are primarily due to biases in short-wave radiation.” My latest understanding is that large biases may exist in the zonal distribution of precipitation over the SO. You can argue that precipitation is part of the LH + SH flux, but when stated as a SW bias, one immediately thinks of supercooled liquid water and radiative transfer, not precipitation. Calling it simply a SW radiation bias is then wrong. I respect, however, that the SW radiation bias is in the literature.

- I have discussed this point with Paul Field, Alejandro Bodas, Kalli Furtado, Keith Williams and John Edwards but unfortunately we do not fully understand what you are suggesting. In the heat flux component sub-section of the drivers of coupled model SST biases section we now assess the AMIP5 net flux and CMIP5 SST biases regression relationship against short-wave, long-wave and total turbulent flux biases. This analysis clearly demonstrates that the largest contribution to AMIP5 net flux and CMIP5 SST bias variations arises from AMIP5 short-wave biases. For example, the very strong correlation of AMIP5 net flux bias on AMIP5 SW bias ($r=0.91$) strongly suggests that AMIP5 short wave biases, which have been linked to AMIP5 cloud biases, are the key control on AMIP5 net flux biases. These analyses also show that the CMIP5 SST and AMIP5 net flux bias on AMIP5 turbulent flux bias correlations are weak. Presumably this relationship would include any precipitation-related contribution to sensible or latent heat? The direct contribution of enthalpy associated with precipitation itself to the total energy flux is also small $<2 \text{ Wm}^{-2}$ (due to the use of a reference temperature in the models) as we state in our Methods (in the sub-section on Multi-model mean and regression analysis). For details see Mayer et al, 2017, reference below which we refer to:

Mayer, M. *et al.* Towards consistent diagnostics of the coupled atmosphere and ocean energy budgets, *J Climate* (2017)

Note also that we find fairly weak relationships between AMIP5 precipitation and SST but a much stronger relationship between CMIP5 and CMIP5-AMIP5 precipitation and SST. This suggests there is a strong feedback of SST biases on Southern Ocean precipitation rather than evidence for precipitation causing SST biases but this is beyond the scope of this study (we would be happy to share and discuss these results with you further after the review).

Please let us know if we have misunderstood this point.

R2.4) If, however, we take this initial leap of faith in the surface fluxes, then model development and conclusions are very interesting.

- Thank you for this encouraging comment.

R2.5) I will state that I 'still' found the primary text difficult to fully appreciate/follow. From the earlier reviews, I note this has been previously discussed and addressed. It was essential for me to read the supplemental material. I respect space is limited, but certainly moving forward some of this material should be considered.

- The theory is not the main focus of the manuscript but was added to provide a plausible physical explanation for our interesting CMIP5 SST bias on AMIP5 net flux bias regression. This is discussed in more detail our initial response letter below. We have already extended the theory detail in the Main text considerably on request from two previous reviews so we are reluctant to extend it too much more. We have agreed with the Editor that we can move the extended theory from Supplementary Material to the Methods so it is more easily accessible and an inherent component of the paper. We have also clarified, and where necessary extended, the theory section of the main text throughout. However, without knowing specific details it is also difficult to know exactly which aspects you would like us to clarify. Please let us know any specific parts of text that you believe still need to be clarified.

3) Initial response letter to the editor: January 9th 2018

Met Office,
Fitzroy Road,
Exeter EX1 3PB.
9th January 2018

Dear Dr Tynan,

Thank you for your response to our manuscript.

We appreciate the reviewers' and your efforts and accept that we need to further improve the manuscript clarity on the new theory. We disagree strongly, however, with your inference from the reviews that this work remains at a preliminary stage. None of the results or main conclusions have changed or will change since the original submission so the revisions from the four reviews have essentially clarified and demonstrated the robustness of the original statements.

The equilibrium mixed layer budget theory was not in our original 2000 word letter submission to Nature Climate Change which received a positive review in Dec 2016 (including Reviewer 1). It was only added simply to address reviewer requests for a plausible physical explanation for our CMIP SST bias on AMIP net flux bias regression result. It is not the main focus of the paper, which is why we opted to include detail in the Supplementary Material. Our aims are therefore to ensure that the theory main text is concise, as well as clear and robust, so as not to detract from the focus of the main manuscript on our results. As we state, if any of our assumptions were not to be valid to a first order then we would not see such a strong correlation ($r=0.84$).

In our opinion, the theory text with its current level of detail adequately fulfils its purpose, i.e. to provide a physical explanation for our interesting regression result. We are therefore reluctant to move too much more material from the Supplementary Material to the Main Text as suggested by Reviewer 2 since it has already been the focus of three sets of review comments, requesting more and more detail. It is also unfortunate that the latest set from Reviewer 1 were not more efficiently raised at either the Mar or Sep 2017 review. One optimal solution to address the reviewer concerns (R1.7; R2.5 below), if permitted, could be to move the theory from the Supplementary Material into an Appendix so that whilst being an integral part of the paper, it does not detract from the main manuscript focus. Prior to submission, we enquired about the use of Appendices in Nature Communications and were informed that it is at the manuscript editor's discretion. Would it therefore be possible to move the Supplementary Theory text into an appendix?

The following substantive new analyses were requested: (a) regression sensitivity studies using a different observational data set and using a model as a reference; (b) calculating the heat flux associated with the mixed-layer temperature tendency (termed H_T) for all of the models; and (c) directly estimating the combined vertical and lateral ocean heat transport convergence, C , term from ocean diagnostics for all the models. (a) is unnecessary since the reviewers have not appreciated that it will make no difference (any regression slopes and correlations are unaffected by parameter references since they are derived from parameter variations with their means subtracted). (b) and (c) would be difficult/impossible to do and would not affect our results or yield new insight (H_T is a small term, $\sim 2\%$ of the STD of the other budget terms, so we could estimate C from $C \sim F$ but we anyway cannot make use of C since we cannot estimate other essential terms, as we state). We could therefore address each of these points by adding a sentence making these points clear. It is likely that Reviewer 1's remaining concern (d) with our causality statements may be related to misunderstandings (b) and (c) above. Note the statement from the original Reviewer 3 who recommended that the paper be accepted in Mar 2017: 'the combined use of AMIP/CMIP experiments is clever and provides very strong novel evidence for a causal link between clouds, radiation, SSTs, and atmospheric circulation over the Southern Ocean'. We could address this comment with one or two modified or extra statements in our theory section summary. We provide detailed evidence on each of these four points, reviewed by named expert co-authors, below this letter.

We are confident we can address all of the other Reviewer comments, and ensure that the manuscript is clear, with some reasonably concise additions and changes to

the existing text. We appreciate the effort of all of the reviewers have made. Given the considerable time-input on all of our parts, we hope that it will be possible to agree what we feel to be a fair and justified way forwards with this. As there is an obvious need for convergence to publication, we are hoping that any further comments could be limited to comments on our new revisions rather new comments on existing material. We would obviously welcome specific suggestions on sections of text that might remain unclear so we can remove, caveat or clarify them. If this suggested approach would be acceptable, we would hope to have these revisions completed within about a month?

Thank you for your help with this matter. We look forward to hearing from you.

Yours sincerely,

Dr Patrick Hyder

Detailed responses to reviewers on the above key points

The following topic-expert co-authors have reviewed and agree with the evidence presented below, and each of our submissions: Jonathan Gregory FRS, Richard Wood and Helene Hewitt (climate and ocean processes/modelling); Richard Allan, Paul Field, John Edwards and Alejandro Bodas (surface exchanges, clouds and atmospheric processes/modelling); and Simon Josey (air-sea exchanges and ocean processes/modelling); Tom Bracegirdle (wind, atmospheric, and sea-ice processes/modelling).

a) Sensitivity of regression results to observational errors - The suggested sensitivity experiments to either using different observational datasets (Reviewer 2, R2.2) or using one of the models as a reference (Reviewer 1, R1.2 and R1.6) would yield identical results and conclusions.

In our introduction we state “Correlations (r), regression slopes (S) and p values (p) do not depend on the accuracy of our observational estimates since the same observational value is subtracted from each of the models. Observational errors only affect the regression intercept values”. This is because for any linear-regression the multi-model mean is subtracted when r and S are estimated. The two parameter means influence only the intercept (in our case for bias estimates these means includes their associated observational errors).

For example, Fig 1 (below) shows the impacts of observational CMIP5 SST and AMIP5 net flux errors on the CMIP5 SST bias on AMIP5 net flux bias regression lines for hypothetical perfectly correlated bias distributions. The correlation and regression results relate to the variations in the SST biases variations across the models to those across the CMIP5 net flux biases (both are exactly known). They do not provide information on multi-model mean biases which obviously do depend on the observational estimates (as we state later in the introduction). We would ensure this is clearer both in the introduction and conclusions.

Fig 1 – The CMIP5 SST bias on AMIP5 net flux bias regression lines and points for hypothetic perfectly correlated SST and net flux bias distributions with a zero intercept (blue line). Also included are the regression line for: (a) a hypothetic correction of 5 Wm^{-2} to the net flux observational estimate (green line); and (b) a hypothetic correction of 0.5K to the SST observational estimate (red line). The correlation and regression slopes are identical for all three series (only the regression intercept changes).

b) Equilibrium assumption in mixed layer theory on CMIP5 SST against AMIP5 net flux regression (Reviewer 1 R1.1) - Analysing the heat budget term associated with the tendency in the mixed layer heat content would be a substantial effort which would be unlikely to significantly alter our results and would not change our conclusions.

The standard deviation of this term for 6 twenty year periods of the HadGEM2 historical run for 40-60°S is 0.16 Wm^{-2} . This is clearly small (~2%) compared to the AMIP net flux bias standard deviation of 6.6 Wm^{-2} . It would remain small even if the models variability were to be a factor of three larger, which on our opinion is unlikely for such a large region. Furthermore, since this term would have acted as an additional noise term in our existing regression analyses, we would not see strong correlations if this term were to be an important term in the budget.

c) Directly estimating C from ocean diagnostics (Reviewer 1 R1.5) – This would be a major task for which suitable diagnostics are likely to be unavailable for most models. If it were possible, it would also yield no extra information or insight.

To estimate the combined vertical and lateral ocean transport convergence into the mixed layer, C, would require special diagnostics averaged online at time-step, e.g. advective, diffusive and parameterised eddy heat transport product terms, which are unlikely to be available for most of the CMIP5 models. This would also be a substantial (~3-6 month task) even for one model and a huge task for many models with differing and irregular spatial and vertical grids.

As we show in point (b), the $\Delta C = -\Delta F$ assumption is to a first order valid so we could estimate ΔC from ΔF . However, as we state, we cannot use our ΔC estimate since there are insufficient ocean-only OMIP5 models to estimate the other necessary term (ΔC_O) to better constrain our estimate the ocean heat transport convergence response sensitivity to local mixed layer temperature, λ_C .

We therefore focus on the relationships between terms we can estimate, as stated, i.e. the CMIP5 SST bias and AMIP5/CMIP5 heat flux bias variations across the models. The unknown ΔC , ΔC_O and ΔC_R terms (and any errors therein) therefore simply act as unknown noise terms in our regressions, limiting the correlations.

We would clarify these points.

d) Validity of statements on causality (Reviewer 1 R1.4) – CMIP5 SST bias variations cannot cause AMIP5 net bias variations since SST is prescribed in the AMIP5 simulations. Hence, as also suggested by the theory, the causality in this relationship can only be that AMIP net flux bias variations across the models cause CMIP5 SST bias variations. This concern may also be related to points (b) and (c) above which we have now clarified.

Note that in the review of our original submission with no theory in Dec 2016 Reviewer 3 stated ‘the combined use of AMIP/CMIP experiments is clever and provides very strong novel evidence for a causal link between clouds, radiation, SSTs, and atmospheric circulation over the Southern Ocean’. On request from Reviewer 1 at the last review we added several paragraphs (line 324-359) to try to ensure the logical step by step chain of causality was clear. We would review and further clarify this text. We state in the methods that our correlations results include cross-correlations. This is particularly true for correlations within CMIP5 (we actually make this point about previous inferences from CMIP5 SST on downward SW correlations). However, our use of CMIP5 against AMIP5 correlations considerably limits the potential candidates that could influence both of our parameters. In our opinion, the only potential candidate for the significant cross-correlations with both parameters would be if local AMIP wind stress parameter biases were correlated with AMIP net flux biases. We investigated this when we initially discovered the strong SST on AMIP net flux correlation and we found them to be extremely weak, as expected for an area mean of such a large area. We could include a specific statement on this in the revisions.

4) Editor reply to our initial response letter: January 9th 2018

Dear Pat,

Thanks for your email.

I have now taken a look at your comments on the peer-review process and appreciate the points you make regarding the required revisions. Your manuscript and proposed revisions will ultimately need to be seen again by our reviewers. Therefore I suggest you revise your manuscript and provide a clear point-by-point to

the reviewer's comments (similarly to what you have done in your letter) that you are happy to return to reviewers.

Regarding your request for moving the theoretical framework to an Appendix, I should say that the only difference between Appendix and Supplementary is in name. Both are outside of the main text and not indexed. Therefore, moving to an Appendix is not an option as it doesn't change anything. For your information, at Nature Communications we have an unlimited Methods section, and upon acceptance of a manuscript, we require all authors to move any methodological information in the Supplementary Information to the main methods section. While not strictly methods, it is partly validation of them and therefore should go in that section. Therefore, I suggest you move the theoretical framework as a new section in the main methods (possibly at the end of that section).

I hope this clarifies things. Please let me know if you have any other questions.

Best,
Eithne

Reviewer #1 (Remarks to the Author):

The authors have made a considerable effort to clarify the assumptions underlying their analysis, the core ideas presented in the manuscript are clearer now, and the caveats are more clearly identified. The authors specifically focus on using CMIP models in combination with AMIP models to identify the role of the atmosphere in generating biases in air-sea heat fluxes. Based on these results, the authors advocate further work on cloud parameterizations. This manuscript provides a useful insights into CMIP and AMIP surface fluxes and offers a framework for future analyses.

I did struggle with syntax in a number of places. In some cases, on-line dictionaries tell me that the phrasing is acceptable in British English (e.g. "in consequence", "in future"), so I can't really argue, since Nature is published in the UK. In other cases, I can't find evidence that the authors' phrasing will be clear to readers.

Points of syntax:

line 87. "convolved in previous analyses". In signal analysis, "convolved" is used to describe applying a filter to data. I don't think the usage here is intended to refer to mathematical convolution, so for clarity, it might be better to say "which were not separable in previous analyses that examined only CMIP5 simulations."

line 152. "on regression figures". My sources say that standard usage is "in figures"

line 347 and following. Discussion of correlation. The methods section explains that the authors have adopted specific terminology: "For linear regressions we employ the terminology 'a correlation of Y on/against X' to refer to evaluating Y as a linear function of X plus errors." That sounds like it should be OK, but there are several problems here. First the authors seem to be using the word "correlation" to refer to "linear regression", which is unconventional and will leave readers thoroughly confused. Maybe I've got the wrong end of the stick, but using "correlation" when you mean "regression" is analogous to saying "In this paper, when we use the word 'blue', we interpret it to mean 'red'." If the discussion is about linear regression, then the text should use the word "regression". Second, this unconventional definition is too far removed from the discussions of correlation in the main text. The definition (if it is retained) should appear in the main body of the text at the point when correlations are first discussed. And third, although the authors indicate that they are using the word "correlation" for "regression", the actual usage, with values of r^2 reported, looks like correlation. If this is correlation, then it would be appropriate to use standard syntax for correlation, and refer to "the correlation of X and Y", or "the correlation between X and Y", or "X

correlates with Y", without using "on" or "against". (I did google this, and I haven't found any statistical documents that use the syntax adopted by these authors.)

line 371. "to be different to"  "to differ from" or "to be different from". (In this case, on-line dictionaries tell me that "different to" is common in British English, but that "different from" is globally more common.

line 412. Change to "would require that ... variations also be driven by"

line 459. "a large fraction of" For clarity, it would help to translate these numbers into a specific fraction (i.e. $1.3/4.8 =$ between a third and a quarter or $\sim 27\%$).

line 462-465. "These estimates are both smaller ... than estimates from bulk formulae of the turbulent flux sensitivity for this region ignoring the atmospheric response of $\geq -20 \text{ W/m}^2/\text{K}$." Is it the atmospheric response or "these estimates" that is $\geq -20 \text{ W/m}^2/\text{K}$?

line 467. "Interestingly" I'd remove this word, as the information in the first sentence seems like an intrinsic property of ocean-only models and not a surprise for readers.

line 503. "much weaker" Weaker than what?

line 1543. "The observed heat content tendency (dH_{obs}/dt) will not affect our regression slope or correlation estimates since its value is the same for all of the models." This statement is not necessarily true, particularly for correlation, which is conventionally normalized by a term that would include dH_{obs}/dt .

line 1624. "models"  "model"

There are 24 authors on this paper, so hopefully several of them have been involved in writing, editing, and reading to try to maximize the clarity and readability of the main manuscript and methods.

Reviewer #2 (Remarks to the Author):

The authors have made substantial revisions to the manuscript. I believe that the new draft is more accessible than the previous one.

With respect to the individual points:

R2.1. This is addressed to my satisfaction. The authors do (repeatedly) acknowledge the large uncertainty in the turbulent flux products and that the spread in various estimates is "considerably larger than the spread between observational estimates of radiative heat fluxes." (lines 963).

Defining these turbulent heat fluxes as a residual is expedient and has been done before.

R2.2. Thank you for clarifying this issue.

R2.3. My apologies for my poor wording. Precipitation does not come into your calculations anywhere. I meant to communicate that the large uncertainty in precipitation reflects large uncertainty in the latent heat flux. And errors in the precipitation in reanalysis products reflect errors in the latent heat flux in reanalysis products.

You already acknowledge large uncertainty in the turbulent fluxes.

Further, this concern is moot in the present configuration/calculation.

R2.5 Again, I acknowledge the authors efforts to make this easier to read and comprehend.

Point by point response to reviewer comments

- We would like to thank both reviewers for their comments which have helped us considerably to improve our manuscript.

a) Reviewer #1:

The authors have made a considerable effort to clarify the assumptions underlying their analysis, the core ideas presented in the manuscript are clearer now, and the caveats are more clearly identified. The authors specifically focus on using CMIP models in combination with AMIP models to identify the role of the atmosphere in generating biases in air-sea heat fluxes. Based on these results, the authors advocate further work on cloud parameterizations. This manuscript provides a useful insights into CMIP and AMIP surface fluxes and offers a framework for future analyses.

- Thank you for this comment. We very much appreciate the considerable amount of time you must have spent on our manuscript over all of the reviews.

I did struggle with syntax in a number of places. In some cases, on-line dictionaries tell me that the phrasing is acceptable in British English (e.g. "in consequence", "in future"), so I can't really argue, since Nature is published in the UK. In other cases, I can't find evidence that the authors' phrasing will be clear to readers.

- Thank you for these comments and your wording suggestions below which were very helpful. Two co-authors and I have re-read the whole manuscript and clarified any remaining issues with syntax/wording.

Points of syntax:

line 87. "convolved in previous analyses". In signal analysis, "convolved" is used to describe applying a filter to data. I don't think the usage here is intended to refer to mathematical convolution, so for clarity, it might be better to say "which were not separable in previous analyses that examined only CMIP5 simulations."

- Done, thanks for this suggestion.

line 152. "on regression figures". My sources say that standard usage is "in figures"

- Done.

line 347 and following. Discussion of correlation. The methods section explains that the authors have adopted specific terminology: "For linear regressions we employ the terminology 'a correlation of Y on/against X' to refer to evaluating Y as a linear function of X plus errors." That sounds like it should be OK, but there are several problems here. First the

authors seem to be using the word "correlation" to refer to "linear regression", which is unconventional and will leave readers thoroughly confused. Maybe I've got the wrong end of the stick, but using "correlation" when you mean "regression" is analogous to saying "In this paper, when we use the word 'blue', we interpret it to mean 'red'." If the discussion is about linear regression, then the text should use the word "regression". Second, this unconventional definition is too far removed from the discussions of correlation in the main text. The definition (if it is retained) should appear in the main body of the text at the point when correlations are first discussed. And third, although the authors indicate that they are using the word "correlation" for "regression", the actual usage, with values of r^2 reported, looks like correlation. If this is correlation, then it would be appropriate to use standard syntax for correlation, and refer to "the correlation of X and Y", or "the correlation between X and Y", or "X correlates with Y", without using "on" or "against". (I did google this, and I haven't found any statistical documents that use the syntax adopted by these authors.)

- Thank you for pointing this out. This was a misunderstanding due to a wording error. We accept that strictly correlation is between X and Y. However, since we often quote correlation values together with the regression slope values which do depend on the direction of the regression analysis we adopted to use this terminology for correlations too. As requested, we have inserted the following text in the main text:

“Most correlation are presented together with regression analyses slopes in brackets or tables. We therefore generally adopt the terminology ‘a correlation of Y on X’ simply to indicate the direction of the associated regression analysis (although the correlation value is between X and Y).”

In the Methods we also state:

“For linear regression analyses we employ the terminology ‘a regression of Y on X’ to refer to evaluating Y as a linear function of X plus errors. Since we mostly quote correlation together with the regression analysis slope we adopt the same terminology for correlation, i.e. we refer to ‘a correlation of Y on X’ simply to indicate the direction of the associated regressions analysis (although correlation values are between X and Y).”

line 371. "to be different to"  "to differ from" or "to be different from". (In this case, on-line dictionaries tell me that "different to" is common in British English, but that "different from" is globally more common.

- Done, thanks again.

line 412. Change to "would require that ... variations also be driven by"

- Done, thanks again.

line 459. "a large fraction of" For clarity, it would help to translate these numbers into a specific fraction (i.e. $1.3/4.8 =$ between a third and a quarter or $\sim 27\%$).

- Done, we have changed this sentence to read ‘around three quarters is re-emitted back towards the surface’. (it is ~73% since the estimated upward long-wave emission is only ~27% of its theoretical value).

line 462-465. "These estimates are both smaller ... than estimates from bulk formulae of the turbulent flux sensitivity for this region ignoring the atmospheric response of ≥ -20 W/m²/K." Is it the atmospheric response or "these estimates" that is ≥ -20 W/m²/K?

- Done, this text has been changed to ‘These estimates are both of considerably smaller magnitude than sensitivities of more than 20Wm⁻²K⁻¹ for the total turbulent flux estimated from bulk formulae for this region ignoring the atmospheric response.’

line 467. "Interestingly" I'd remove this word, as the information in the first sentence seems like an intrinsic property of ocean-only models and not a surprise for readers.

- Done.

line 503. "much weaker" Weaker than what?

- Done, we have changed the wording to “However, AMIP5 ZWML biases are only weakly anti-correlated with AMIP5 net flux biases ($r=-0.44$).”

line 1543. "The observed heat content tendency (dH_{obs}/dt) will not affect our regression slope or correlation estimates since its value is the same for all of the models." This statement is not necessarily true, particularly for correlation, which is conventionally normalized by a term that would include dH_{obs}/dt .

- For our known variables, i.e. SST biases and AMIP5 net flux biases, adding a constant to either variable distribution makes no difference to correlation or slope values (as it does not affect the variances as we explained and demonstrated for the observational errors). We have therefore not changed the wording.

- It is possible that the confusion has arisen since if one models all of the terms on the right hand side as Gaussian distributions, including the $(\lambda_C + \lambda_F)$ denominator, and then calculate the SST bias, then adding an extra constant error term, changes the SST distribution and therefore changes the correlation (it has no impact if the $(\lambda_C + \lambda_F)$ denominator is assumed to be constant). However, in our case the SST distribution is known and therefore cannot change when the extra term is added (i.e. any right hand side error terms are already included in the SST on net flux regression analysis residuals).

line 1624. "models"  "model"

- Done, thanks.

There are 24 authors on this paper, so hopefully several of them have been involved in writing, editing, and reading to try to maximize the clarity and readability of the main manuscript and methods.

- Two co-authors and I have re-read the manuscript and improved any remaining issues with syntax (the manuscript has been read by several co-authors before every submission).

Reviewer #2 (Remarks to the Author):

The authors have made substantial revisions to the manuscript. I believe that the new draft is more accessible than the previous one.

- Thanks you for this comment.

With respect to the individual points:

R2.1. This is addressed to my satisfaction. The authors do (repeatedly) acknowledge the large uncertainty in the turbulent flux products and that the spread in various estimates is "considerably larger than the spread between observational estimates of radiative heat fluxes." (lines 963).

Defining these turbulent heat fluxes as a residual is expedient and has been done before.

- Thank for you this comment. The repetition in the Methods was unavoidable to introduce the subsequent sentence where we quote the difference values (which we did not want to include in the main text). However, we have changed this text slightly to read: "As discussed, this approach is adopted since the spread in observational estimates of turbulent fluxes is considerably larger than the spread between observational estimates of radiative heat fluxes. For example, for 40-60°S the OAFLUX versus SEAFLUX total turbulent flux difference is 25Wm^{-2} and the CERES-EBAF versus ISCCP total radiative flux difference is 6Wm^{-2} ."

R2.2. Thank you for clarifying this issue.

- Thanks.

R2.3. My apologies for my poor wording. Precipitation does not come into your calculations anywhere. I meant to communicate that the large uncertainty in precipitation reflects large uncertainty in the latent heat flux. And errors in the precipitation in reanalysis products reflect errors in the latent heat flux in reanalysis products. You already acknowledge large uncertainty in the turbulent fluxes. Further, this concern is moot in the present configuration/calculation.

- Thank you, this comment was useful to make us think more about the relationship between precipitation and SST biases, which is an interesting topic.

R2.5 Again, I acknowledge the authors efforts to make this easier to read and comprehend.

- Thanks again.